# Oxygen levels at the time of activation determine T cell persistence and immunotherapeutic efficacy

Pedro P Cunha[1], Eleanor Minogue[1], Lena CM Krause[1,2], Rita M Hess[1,3], David Bargiela[1], Brennan J Wadsworth[4], Laura Barbieri[1,4], Carolin Brombach[4], Iosifina P Foskolou[1], Ivan Bogeski[2], Pedro Velica[4], Randall S Johnson[1,4]*

[1]Department of Physiology, Development and Neuroscience, University of Cambridge, Cambridge, United Kingdom; [2]Molecular Physiology, Institute of Cardiovascular Physiology, University Medical Center, Georg-August-University, Göttingen, Germany; [3]Cancer Research UK, Cambridge Institute, University of Cambridge, Cambridge, United Kingdom; [4]Department of Cell and Molecular Biology, Karolinska Institute, Stockholm, Sweden

**Abstract** Oxygenation levels are a determinative factor in T cell function. Here, we describe how oxygen tensions sensed by mouse and human T cells at the moment of activation act to persistently modulate both differentiation and function. We found that in a protocol of CAR-T cell generation, 24 hr of low oxygen levels during initial CD8[+] T cell priming is sufficient to enhance antitumour cytotoxicity in a preclinical model. This is the case even when CAR-T cells are subsequently cultured under high oxygen tensions prior to adoptive transfer. Increased hypoxia-inducible transcription factor (HIF) expression was able to alter T cell fate in a similar manner to exposure to low oxygen tensions; however, only a controlled or temporary increase in HIF signalling was able to consistently improve cytotoxic function of T cells. These data show that oxygenation levels during and immediately after T cell activation play an essential role in regulating T cell function.

## Editor's evaluation

Your work employing solid preclinical models and in vitro experiments, and addressing how oxygen tension affects T-cell function, can help to improve cellular therapies for future patients.

*For correspondence:
rsj33@cam.ac.uk

Competing interest: The authors declare that no competing interests exist.

## Introduction

Cytotoxic T cells are often faced with oxygen-poor conditions (*Labani-Motlagh et al., 2020*), and in this context, tissue oxygen scarcity (hypoxia) is known to contribute to immune tolerance in tumours (*Noman et al., 2015*). For example, tumour hypoxia recruits immunosuppressive regulatory T cells, which can inhibit CD8[+] T cell function (*Facciabene et al., 2011*). Hypoxia is also known to decrease expression of MHC class I molecules in tumour cells themselves, which in turn hampers T cell priming and cytotoxic function (*Sethumadhavan et al., 2017*). Tumour hypoxia can also have direct inhibitory effects on CD8[+] T cells by increasing the expression of checkpoint inhibitor receptors (*Bannoud et al., 2021*; *Noman et al., 2015*) and cause mitochondrial dysfunction (*Liu et al., 2020*; *Scharping et al., 2021*). However, T cells infiltrate a wide variety of tissues throughout their progression to maturation and final differentiation; they therefore need to function within a wide range of oxygen partial pressures, including oxygenation levels that border on anoxia in some lymphoid organs (*Caldwell et al.,*

*2001*). Thus, oxygen-sensing mechanisms in T cells must be finely tuned for proper functioning in various tissue microenvironments.

In both preclinical studies and clinical manufacturing, CD8$^+$ T cells are typically cultured at atmospheric sea-level $O_2$ conditions (approximately 20.5%); however, reports have shown that physiologically relevant oxygen tensions (corresponding to 1–5% $O_2$) can profoundly shift T cell differentiation and function (*Atkuri et al., 2007*; *Caldwell et al., 2001*; *Gropper et al., 2017*; *Larbi et al., 2010*; *Loeffler et al., 1990*; *Naldini et al., 1997*; *Palazon et al., 2017*; *Ross et al., 2021*; *Veliça et al., 2021*). Lower oxygen levels alter CD8$^+$ T cell effector differentiation in a process that requires hypoxia-inducible factors (HIFs), the key transcriptional regulators of cellular responses to low oxygen (*Finlay et al., 2012*; *Palazon et al., 2017*).

Culturing CD8$^+$ T cells ex vivo at 1% $O_2$ has been found to dramatically reduce T cell proliferation, but increase antitumour activity, upon transfer into tumour-bearing mice (*Gropper et al., 2017*). Reduced oxygenation and increased HIF-α levels can thus both impair and enhance CD8$^+$ T cell cytotoxicity. We hypothesised that this might be related to the timing of hypoxic stress during T cell activation and differentiation.

In this study, by combining the use of low oxygen conditioning with pharmacological inhibition of negative HIF regulators, we show that hypoxia has an unexpected role at the point of activation and for a short period thereafter. The effects of short hypoxic conditioning are highly persistent, eliciting both ex vivo and in vivo increases in effector T cell differentiation and antitumour efficacy, and demonstrate the critical role oxygen plays in the earliest stages of T cell activation.

## Results

### Hypoxia and increased HIF signalling drive CD8$^+$ T cell effector differentiation

To assess the effect of different oxygen tensions in T cells, we first isolated polyclonal CD8$^+$ T cells from C57BL/6J mice that were activated with anti-CD3 and anti-CD28 dynabeads for 3 d in ambient oxygen (21%) and 5 or 1% $O_2$. Low oxygen levels increased effector differentiation of T cells, upregulating molecules such as the IL-2 receptor alpha subunit CD25, the transcription factors EOMES, T-bet, and TOX, and the effector proteins granzyme B (GZMB) and interferon-γ (IFN-γ) (*Figure 1A*). T cells cultured at 1% $O_2$ have increased expression of inhibitory receptors and impaired clonal expansion (*Figure 1B* and *Figure 1—figure supplement 1A*). The HIF transcription factors are one of the primary mechanisms by which T cells respond to hypoxia and are critical to T cell function (*Palazon et al., 2017*). Contrary to many cell types, an increase in HIF-1α mRNA and protein levels can occur in T cells even at high oxygen levels, following stimulation of the T cell receptor (TCR) (*Finlay et al., 2012*; *Lukashev et al., 2006*; *Makino et al., 2003*; *Nakamura et al., 2005*; *Wang et al., 2011*). We first wished to determine whether the kinetics of HIF-1α stabilisation in ex vivo activated CD8$^+$ T cells is dependant on surrounding oxygen tensions (*Figure 1C*). In all oxygen tensions tested, HIF-1α protein expression increased after T cell activation and peaked within 12 hr. Exposure to 1% $O_2$ resulted in the greatest accumulation of HIF-1α during the 3 d of culture (*Figure 1C*). HIF-2α protein was also increased in T cells following activation, but was mostly found in the cytoplasmic protein fraction (*Figure 1—figure supplement 1B*), as previously seen (*Park et al., 2003*). This is consistent with data showing that deletion of HIF-1α, but not HIF-2α, abrogates hypoxia-driven increases in GZMB (*Figure 1—figure supplement 1C*) and other activation markers, and impairs antitumour T cell function (*Palazon et al., 2017*).

We next wished to determine which of the hypoxia-driven alterations in T cell activation rely on the observed increase in HIF-1α levels elicited by low oxygen tensions (*Figure 1C*). We did this by increasing HIF signalling through pharmacological inhibition of prolyl hydroxylation, utilising the prolyl hydroxylase inhibitor FG-4592 (*Chen et al., 2019*; *Figure 1D*). Both low oxygen tensions and FG-4592 act to inhibit proteasomal degradation of HIF-α, leading to HIF-1α accumulation (*Figure 1E*). To facilitate functional characterisation, we used CD8$^+$ OT-I T cells derived from mice carrying a transgene to express a specific TCR that reacts to the SIINFEKL peptide of ovalbumin (OVA) (*Hogquist et al., 1994*) Similarly to polyclonal T cells, low oxygen conditioning of OT-I cells for 3 d following activation impaired clonal expansion, and increased levels of CD44, CD127, GZMB, and IFN-γ (*Figure 1F* and *Figure 1—figure supplement 1D*). At the selected concentration (50 μM), FG-4592 increased the

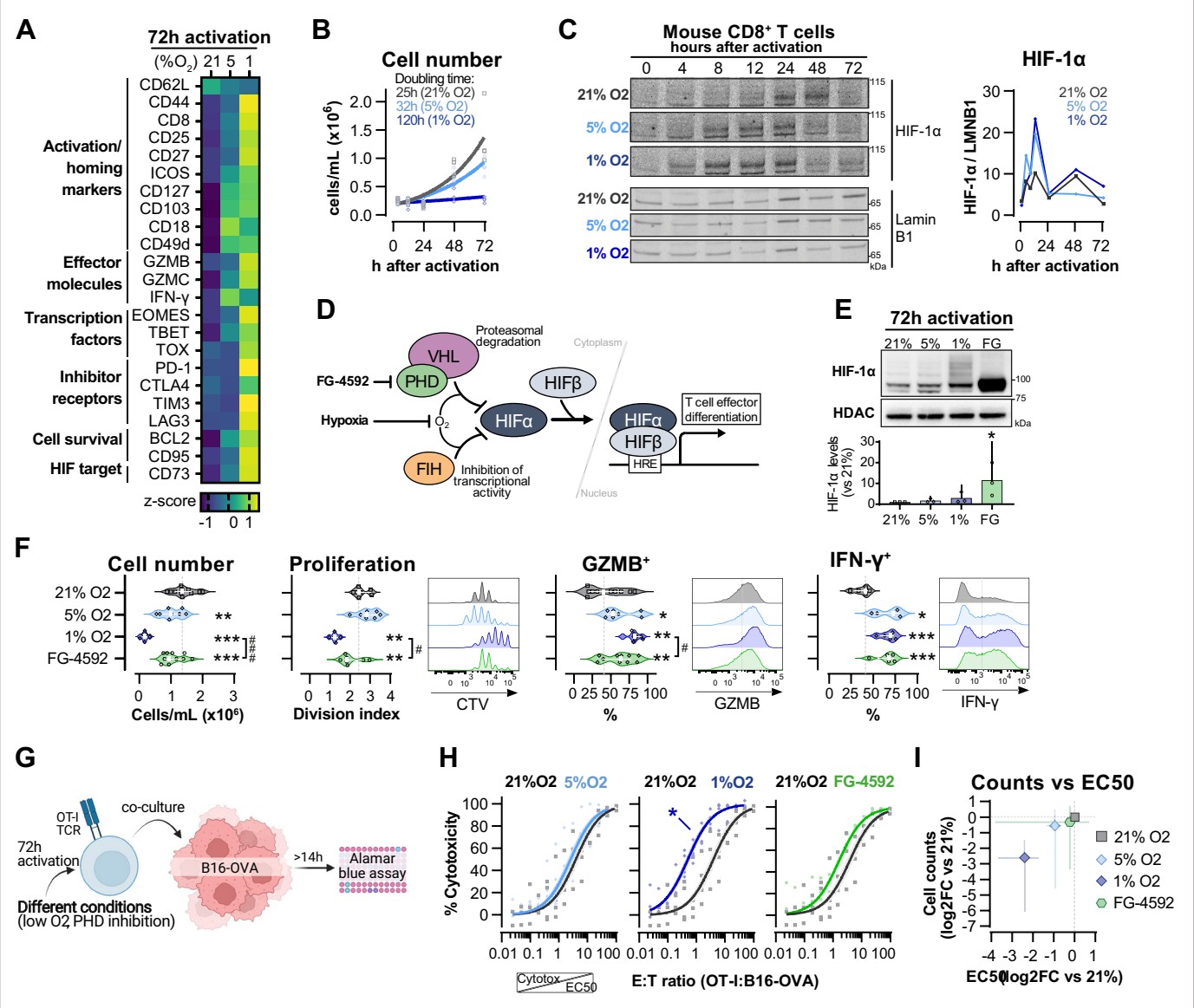

**Figure 1.** Hypoxia and inhibition of negative hypoxia-inducible factor (HIF) signalling in CD8+ T cell effector differentiation. (**A**) Heat map illustrating expression of markers of differentiation as median fluorescence intensity (MFI) determined by flow cytometry 72 hr after activation of naive CD8+ T cells in 21, 5, or 1% $O_2$. Viridis was used as colour scale, and rows represent averaged z-scores; n = 4–29. (**B**) CD8+ T cell expansion following activation in 21, 5, or 1% $O_2$ determined with automated cell counter. Exponential growth curve fit (represented by thick lines) was used to calculate doubling time in hours of T cells in each oxygen tension; n = 4. (**C**) HIF-1α protein expression in nuclear extracts from CD8+ T cells activated in 21, 5, or 1% $O_2$. Representative immunoblots (left) and HIF-1α protein signal normalised to Lamin B1 (right); n = 1. (**D**) Representation of inhibition of negative regulators of HIFα with hypoxia and chemical inhibition of PHD with FG-4592. (**E**) HIF-1α protein expression in nuclear extracts from OT-I CD8+ T cells activated for 72 hr. Conditions analysed: wild-type (WT) cells activated in 21, 5, or 1% $O_2$; WT cells activated in 21% $O_2$ and treated with 50 μM FG-4592. Representative immunoblot (top) and HIF-1α levels relative to 21% after normalisation to HDAC or Histone 3 loading controls (bottom); n = 3. (**F**) Analysis of cell number, proliferation, and expression of differentiation markers in the experimental con0ditions described in (**E**). Cell number determined with automated cell counter and cell proliferation with CTV staining. Expression of differentiation markers was measured after stimulation with SIINFEKL and brefeldin for 3 hr and is shown as a percentage of live CD8+ cells. Histograms are representative flow cytometry plots for each parameter and are pre-gated on live, singlet, CD8+ events. The dotted line defines in violin plot graphs the median of 21% grown cells and in histograms the gate for each marker; n = 8–30. (**G**) In vitro cytotoxicity assay using OT-I CD8+ T cells shown in (**F**). OT-I cells were co-cultured with B16–OVA tumour cells at different effector:target (E:T) ratios and cytotoxicity was assessed after 14–18 hr. (**H**) Cytotoxicity assay obtained according to (**G**) performed at 1% $O_2$ and shown in dose–response curves (plotted with 95% confidence intervals as shaded areas) determined with non-linear regression ([agonist] vs. normalised response). Asterisk represents significantly lower $EC_{50}$ values compared to 21%; n = 4–7. (**I**) Correlation between $log_2$ fold change (FC) of cell counts on day 3 (obtained in **F**) and $EC_{50}$ values (obtained in **H**) relative to 21% $O_2$. All results were pooled from at least two

*Figure 1 continued on next page*

*Figure 1 continued*

independent experiments and are shown as median ± interquartile range (IQR). Each data point represents an independent animal. *p<0.05, **p<0.01, ***p<0.001; Holm–Šídák's multiple-comparisons test relative to 21% (**E–H**) or #p<0.05, ###p<0.001; paired *t* test (**F**).

The online version of this article includes the following source data and figure supplement(s) for figure 1:

**Source data 1.** Raw data and detailed analysis for *Figure 1*.

**Figure supplement 1.** Effects of hypoxia-inducible factor (HIF) stabilisation on CD8+ T cell viability, differentiation, and cytotoxicity at different $O_2$ tensions.

**Figure supplement 1—source data 1.** Raw data and detailed analysis for *Figure 1—figure supplement 1*.

expression of the effector molecules GZMB and IFN-γ and had a reduced impact on cell expansion when compared to 1% $O_2$, which is relevant for an ex vivo protocol of T cell expansion (*Figure 1F*). Importantly, expression of GZMB and IFN-γ was assessed following SIINFEKL rechallenge at day 3, showing that both low oxygen conditioning and FG-4592 could prime T cells with an enhanced capacity to respond to restimulation.

OT-I cells grown in 1% $O_2$ showed increased antigen-specific cytotoxicity against B16F10 melanoma tumour cells expressing OVA peptide when compared to wild-type CD8+ T cells grown at 21% $O_2$ (*Figure 1G and H*). This increased cytotoxicity occurred when assays were conducted at 1% $O_2$, 5% $O_2$, or 21% $O_2$ (*Figure 1—figure supplement 1E*). Despite belonging to the physiological oxygen range at which cells are exposed to in vivo, 5% $O_2$ did not impact OT-I cell function under our experimental conditions in vitro, nor did the 3 d treatment with FG-4592 (*Figure 1H*). This might both indicate that only a more pronounced reduction in oxygen levels can functionally modulate T cells, and that the effects of low oxygen conditioning are not fully driven by altered HIF signalling.

Increased cytotoxicity in these populations (illustrated by lower $EC_{50}$ values) was accompanied by decreased cell numbers on day 3 (*Figure 1I*), highlighting that a trade-off relationship exists between CD8+ T cell proliferation and effector function when these are induced by low oxygen tensions and increased HIF signalling.

## Modulation of hypoxic response ex vivo improves antitumour T cell function in vivo

We next compared the effects of hypoxia and HIF signalling modulation on antitumour CD8+ T cell function by employing a model of adoptive cell transfer (ACT) (*Figure 2A*). This model allows for an assessment of the long-term effects of temporary exposure of CD8+ T cells to low oxygen or PHD inhibition. Wild-type CD45.2+ mice were subcutaneously inoculated with OVA-expressing B16 cells, lymphodepleted with cyclophosphamide, and then injected with CD45.1+ OT-I CD8+ T cells after 3 d of activation. The use of the CD45.1 and CD45.2 markers allowed us to follow transfused cells after adoptive transfer into experimental recipients (*Figure 2—figure supplement 1A*).

Culturing of OT-I CD8+ T cells in 1% $O_2$ ex vivo markedly improved expansion in peripheral blood when the numbers of transferred cells were assayed at 7 d after ACT (*Figure 2B and C*). These results, whereby culture of T cells ex vivo in hypoxia gives rise to a striking in vivo proliferative advantage, were quite unexpected. Cultivation of T cells ex vivo showed that hypoxic stress (i.e., culture in low oxygen) greatly reduces proliferation following activation (*Figure 1B and F*). T cells cultured in 1%$O_2$ additionally had a higher proportion of terminally differentiated CD62L-CD44+ cells 7 d after ACT (*Figure 2D*), following their pattern of increased expression of CD44 and reduced levels of CD62L observed in vitro (*Figure 1A*). ACT of OT-I cells cultured in 5% $O_2$, or in the presence of FG-4592, did not impact their expansion in the blood or their expression of CD44 or CD62L (*Figure 2B–D*).

Tumour growth was measured for up to 40 d following ACT. The transfer of any OT-I cells delayed tumour growth in comparison with animals not receiving ACT (*Figure 2E–G*). When compared to animals receiving WT OT-I cells grown at 21% $O_2$, adoptive transfer of FG-4592- or 1% $O_2$-treated OT-I T cells was found to initially decrease tumour burdens (*Figure 2E and F*); while FG-4592-treated cells significantly extended animal survival (*Figure 2G*). Animals receiving OT-I cells conditioned to 5% $O_2$ in vitro did not show differences relative to control tumour growth (*Figure 2E–G*).

These data show that enhanced effector function of T cells conditioned to low oxygen, or to PHD inhibition ex vivo translates to an improved antitumour efficacy in vivo following ACT.

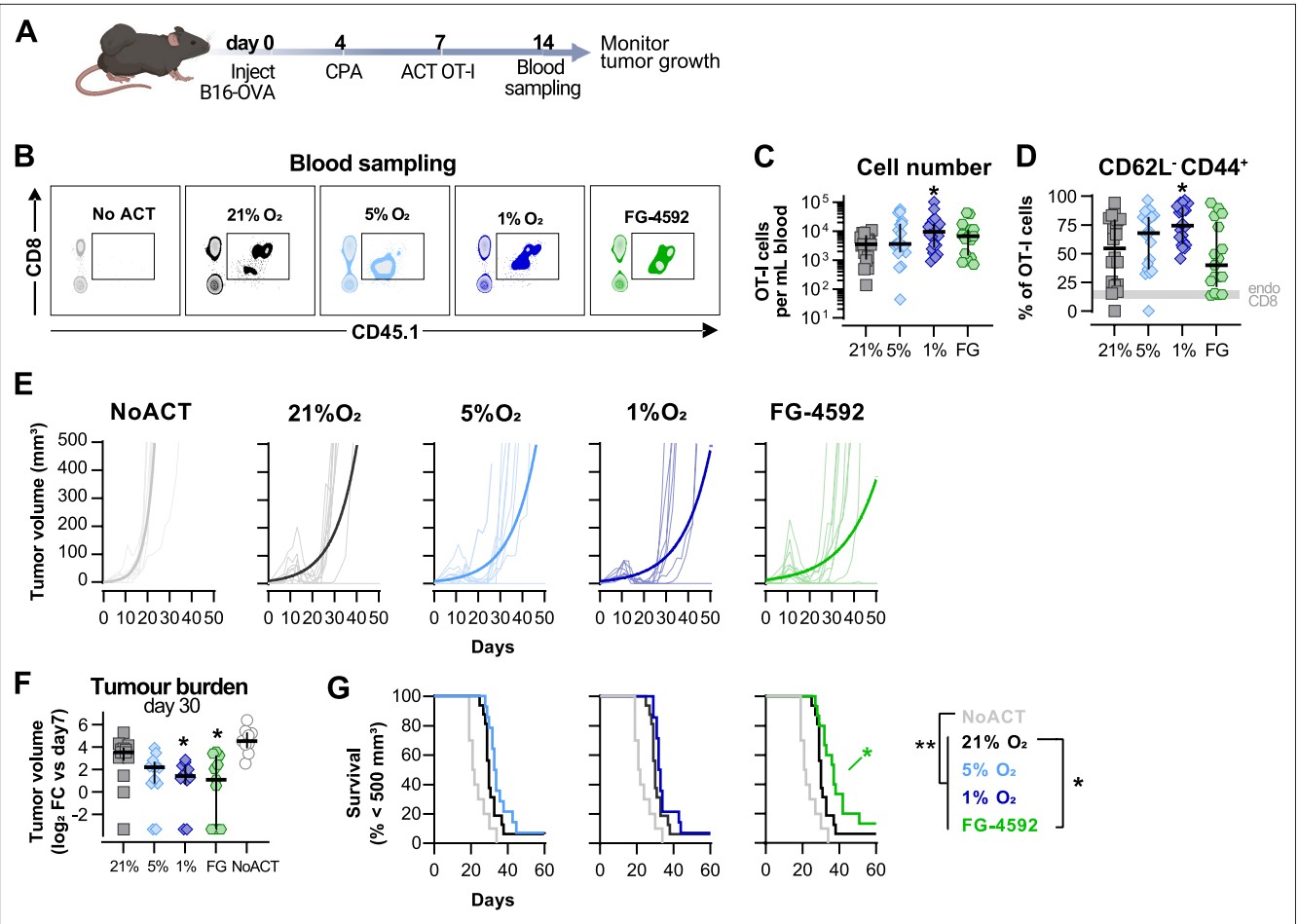

**Figure 2.** Effects of hypoxia and increased hypoxia-inducible factor (HIF) signalling in antitumour function of CD8+ T cells. (**A**) Model of adoptive cell transfer (ACT) therapy. CD45.2+ C57BL/6j mice were inoculated with B16-OVA cells, lymphodepleted with cyclophosphamide (CPA) and injected with CD45.1+ OT-I CD8+T cells activated 3 d prior. Peripheral blood was sampled and analysed by flow cytometry. Tumour growth was monitored every 2–3 days until day 60. (**B**) Blood analysis on day 14. Representative flow cytometry plots with events pre-gated on live, singlet, CD45+. (**C, D**) Frequency of adoptively transferred OT-I cells per millilitre of peripheral blood (**C**) and percentage of CD62L-CD44+ among CD8+ CD45.1+ in peripheral blood (**D**) on day 7 after ACT; n = 17–20, median ± IQR. (**E**) Tumour growth curves. Animals not receiving T cells (NoACT) were used as negative controls. Thin lines: individual animals; thick lines: Malthusian growth curve fit; n = 7–10. (**F**) Tumour burden on day 30 shown as $\log_2$ fold change relative to day 7 (day of ACT). The value of 0.1 (corresponding to the smallest $\log_2$ fold change detected) was added to all conditions to allow the plotting of animals with no tumours; n = 9–14, median ± IQR. (**G**) Survival curves for tumour growth shown in (**D**). Threshold for survival was set to 500 mm³; n = 10–16. All results (except **E**) were pooled from two independent experiments and each data point represents an independent animal. *p<0.05, **p<0.01, ***p<0.001; Kruskal–Wallis test relative to 21% corrected with Dunn's test (**C, D, F**) and log-rank (Mantel–Cox) test relative to 21% $O_2$ or NoACT groups (**G**).

The online version of this article includes the following source data and figure supplement(s) for figure 2:

**Source data 1.** Raw data and detailed analysis for *Figure 2*.

**Figure supplement 1.** (**A**) Gating strategy for analysis of peripheral blood.

## Fine-tuned HIF signalling improves CAR-T cell cytotoxicity against solid tumours

Results shown in *Figures 1 and 2* argue that an increase in HIF signalling through PHD inhibition with FG-4592 can act to improve cytotoxic T cell function in a similar manner to that observed with low oxygen conditioning. To determine whether increased HIF signalling can improve T cell function in vivo in CAR therapies and in human T cells, we employed a model of CAR-T cell therapy. We compared the effects of pharmacological inhibition of HIF degradation using FG-4592 ex vivo to the constitutive downregulation of HIF degradation using a vector construct to silence the E3 ubiquitin ligase VHL (*Figure 3A*). For this, we used HER2 CAR-T cells targeted to a solid tumour model of SKOV3 human ovarian epithelial cancer. RQR8 was used as a transduction marker after confirming

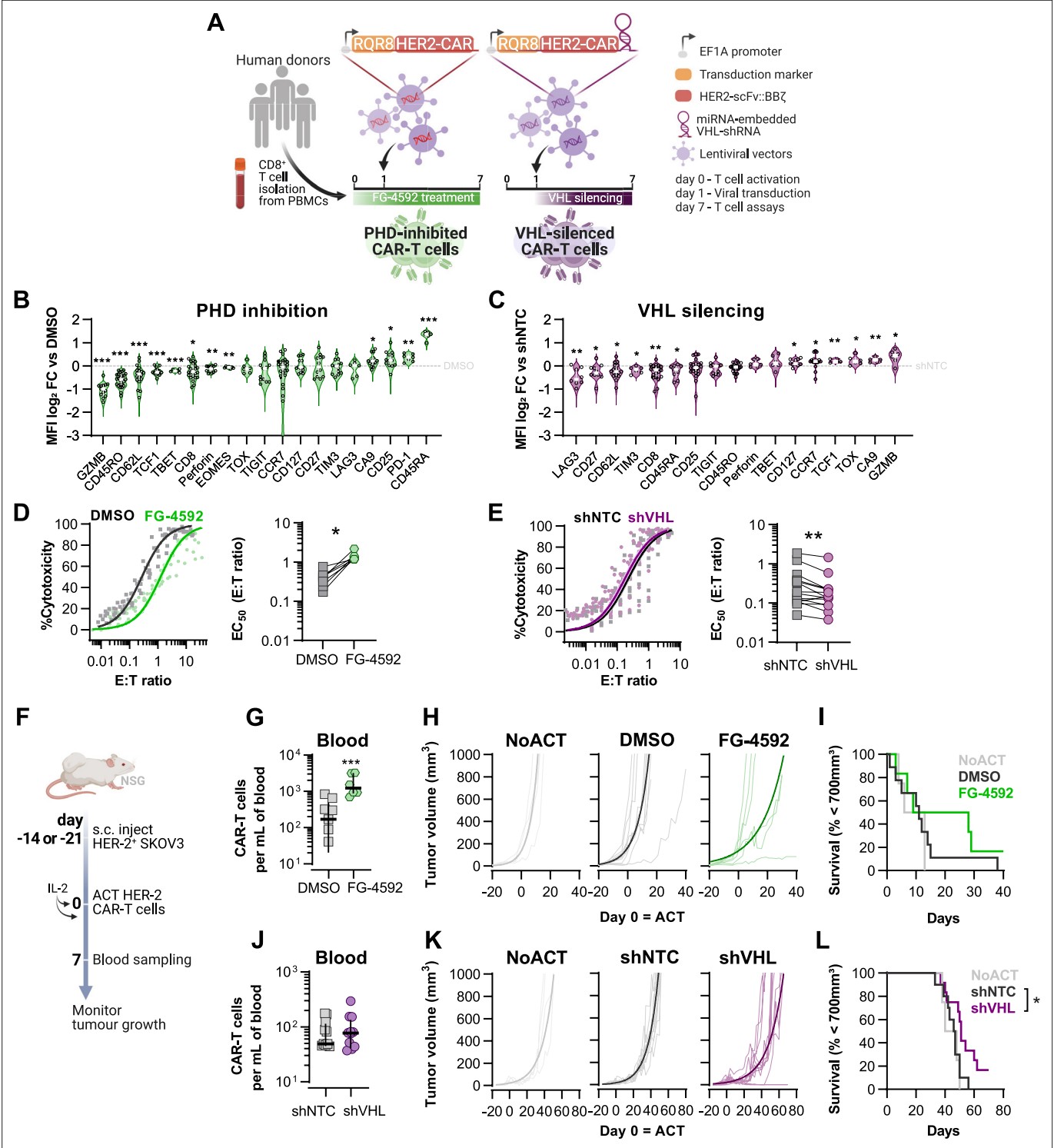

**Figure 3.** Effects of modified hypoxia-inducible factor (HIF) signalling in CART cell cytotoxicity against solid tumours. (**A**) Generation of CAR-T cells. Human CD8[+] T cells were activated, transduced HER2-CAR vectors the day after and assayed on day 7. To modulate HIF signalling, 50 μM FG-4592 (FG) were added to CAR-T cell cultures or cells were transduced with a HER2-CAR vector containing a miRNA-embedded shRNA against VHL (shVHL). DMSO treatment and transduction with CAR vector containing a non-targeted control shRNA (shNTC) were used as controls for FG and shVHL conditions, respectively. RQR8 was used as transduction marker. (**B, C**) Flow cytometry analysis of differentiation markers relative to each respective control (horizontal grey line) following PHD inhibition with FG treatment (**B**) or VHL silencing (**D**); n = 6–30, median ± IQR. (**D, E**) In vitro cytotoxicity assay. DMSO- or FG-treated CAR-T cells (**D**) and shNTC- or shVHL-expressing CAR-T cells (**E**) were co-cultured with SKOV3 tumour cells at different effector:target (E:T) ratios and cytotoxicity was assessed after 14–18 hr of co-culture at 1% $O_2$. Left: dose–response curves (plotted with 95% confidence

*Figure 3 continued on next page*

*Figure 3 continued*

intervals represented in shaded areas) determined with non-linear regression ([agonist] vs. normalised response); right: $EC_{50}$ values obtained from non-linear regression; n = 7–14. (**F**) Model of CAR-T cell therapy. NOD.Cg-*Prkdc^scid^Il2rg^tm1Wjl^*/SzJ mice were inoculated with SKOV3 cells and injected with CAR-T cells generated according to (**A**). Peripheral blood was sampled and analysed by flow cytometry. Tumour growth was monitored every 2–3 d until day 40 or 70 for FG and shVHL experiments, respectively. (**G, J**) Frequency of adoptively transferred CAR-T cells per millilitre of peripheral blood on day 7 for DMSO- or FG treatment (**G**) and shNTC- or shVHL-silencing (**J**); n = 6-12, median ± IQR. (**H, K**) Tumour growth curves experiment with DMSO- or FG-treated CAR-T cells (**H**) and shNTC- or shVHL-silenced CAR-T cells (**K**). Animals not receiving T cells (NoACT) were used as negative controls in both experiments. Thin lines: individual animals; thick lines: Malthusian growth curve fit; n = 5-12. (**I, L**) Survival curves for tumour growth shown in (**H**) and (**K**). Threshold for survival was set at 700 mm³. All results (except **G–L**) were pooled from two independent experiments, and each data point represents an independent animal. *p<0.05, **p<0.01, ***p<0.001; one-sample *t* test relative to control (**B, C**), Wilcoxon matched-pairs signed-rank test (**D, E**), unpaired *t* test (**G**), and log-rank (Mantel–Cox) test relative to shNTC (**L**).

The online version of this article includes the following source data and figure supplement(s) for figure 3:

**Source data 1.** Raw data and detailed analysis for *Figure 3*.

**Figure supplement 1.** Validation of shRNA-CAR vectors and complementary analysis of FG-treated and VHL-silenced T cells.

**Figure supplement 1—source data 1.** Raw data and detailed analysis for *Figure 3—figure supplement 1*.

co-expression with CAR molecules at the surface (*Figure 3—figure supplement 1A*). To achieve VHL silencing, we engineered a CAR vector that is coupled to the expression of a microRNA-embedded Vhl shRNA (*Figure 3A*). In a proof-of-concept experiment, CD5 (highly expressed in lymphocytes) was successfully downregulated in T cells transduced with CD5-shRNA vectors, as confirmed by flow cytometry in RQR8+ cells, when compared to RQR8- events (non-transduced) or with cells transduced with non-targeted control (NTC) shRNAs (*Figure 3—figure supplement 1A and B*). This occurred whether the vector co-expressed a HER2-CAR sequence along with the shRNA or not. Silencing of CD5 expression was stable until at least day 12 after transduction (*Figure 3—figure supplement 1C*).

Transduction of human CD8+ T cells with VHL-targeted shRNA (shVHL) using an engineered CAR vector reduced VHL expression by approximately 30% (*Figure 3—figure supplement 1D*); this was sufficient to detectably increase HIF levels (*Figure 3—figure supplement 1E*). Culturing T cells for 7 d with PHD inhibition, or silencing VHL by shRNA, increased expression of the HIF target gene carbonic anhydrase 9 (Ca9) and reduced expression of CD62L relative to non-treated controls (*Figure 3B and C*). However, expression of most markers of differentiation was remarkably different between these two cell populations and relative to controls. In particular, GZMB, TOX, and TCF1 were decreased by pharmacological PHD inhibition, but were increased in T cells transduced with the VHL silencing vector. FG-4592 treatment impaired basal and maximum oxygen consumption rates of T cells, and VHL silencing with this vector did not significantly impact T cell metabolism, as assessed by a mitochondrial stress test (*Figure 3—figure supplement 1F and G*). In vitro, while FG-4592 treatment reduced cytotoxicity of CAR-T cells, shVHL-expressing CAR-T cells showed enhanced tumour cell killing capacity in 21 and 1% $O_2$ (*Figure 3D and E* and *Figure 3—figure supplement 1H and I*). In contrast to 7 d treatments, short-term exposure to FG-4592 for 1 or 3 d did not impair antitumour T cell function (*Figure 3—figure supplement 1J and K*), implying that the timing of HIF expression during T cell activation is crucial for regulation of cytotoxicity.

We next employed a model of ACT using these HER2-CAR-T transduced human T cells. This allowed for an assessment of long-term effects of both temporary and constitutive modulation of the HIF pathway achieved through the pharmacological FG-4592 treatment ex vivo or via the vector-induced silencing of VHL, respectively. Immunocompromised (NSG) mice were subcutaneously inoculated with HER2-expressing SKOV3 human cancer cells and then injected with RQR8+ HER2-CAR-expressing human donor CD8+ T cells, generated as shown in *Figure 3A*. After ACT was delivered to recipient animals, FG-treated CAR-T cells expanded to a greater extent in the blood relative to DMSO-treated control T cells (*Figure 3G*). Tumour growth analyses showed that, relative to animals not receiving CAR-T cells, FG-4592-treated CAR-T cells delayed tumour growth in 50% of the animals compared to 9% of the animals in the control T cell group (*Figure 3H*). However, the FG-4592 treatment group did not statistically significantly improve animal survival in spite of having one animal with a complete cure (*Figure 3I*). VHL silencing by the shRNA vector did not impact cell expansion in vivo (*Figure 3J*). However, VHL-silenced HER2-CAR-T cells were able to decrease tumour growth significantly, and an improved animal survival relative to NTC-expressing CAR-T cells was seen, clearing tumours in 2 of the 11 animals analysed (*Figure 3K and L*).

These data show that an increase in HIF signalling can potentiate T cell responses in vivo, as seen both by increased expansion of FG-treated cells and through increased tumour killing capacity of VHL-silenced cells, indicating that HIF expression can therapeutically increase human CAR-T cell function.

## Short hypoxic conditioning is sufficient to shape T cell differentiation

We found that low oxygen conditioning increases HIF stabilisation in activated CD8$^+$ T cells (*Figure 1C*). After establishing that protocols of HIF stabilisation in T cells with PHD inhibition or VHL silencing can modulate differentiation and function, we became interested in understanding whether the duration of low oxygen exposure would also have functional consequences for CD8$^+$ T cells. For this, we conditioned human CD8$^+$ T cells for 1, 3, or 7 d at 1% $O_2$ during a 7 d culture protocol and compared their differentiation to that of cells cultured exclusively under ambient (21% $O_2$) conditions (CT) (*Figure 4A*).

As was seen in murine CD8$^+$ T cells, incubation in low oxygen conditions decreases human CD8$^+$ T cell expansion (*Figure 4—figure supplement 1A*). Longer periods spent at 1% $O_2$ correlated with the strongest reduction in cell numbers when compared to 21% $O_2$-grown cells (*Figure 4B*). However, the expression of differentiation markers elicited by 1, 3, and 7 d of 1% $O_2$ conditioning was remarkably similar, although the magnitude of differential expression increased with longer exposure to 1% $O_2$ (*Figure 4C* and *Figure 4—figure supplement 1B*). A similar trend was found when we carried out exposures to 5% $O_2$ conditioning (*Figure 4—figure supplement 1C and D*). On the other hand, incubation in low oxygen conditions (1 or 5% $O_2$) during the last 3 d of culture had little effect (*Figure 4—figure supplement 1E and F*), despite leading to the highest levels of HIF-1α on day 7 (*Figure 4—figure supplement 1G*). Conditioning of T cells in 1% $O_2$ for the first 24 hr of activation (1% 1 d) increased both day 7 proportions of CD45RO$^+$CCR7$^-$ cells and the expression of the effector molecule perforin, and did this regardless of how many times cell culture media was changed during the ambient oxygen expansion (*Figure 4—figure supplement 1H*).

In fact, a single day of 1% $O_2$ conditioning, occurring at the time of activation, followed by 6 d of 21% $O_2$ culture, was sufficient to elicit a classical hypoxia/HIF-driven metabolic adaptation in T cells as characterised by an increased baseline glycolytic rate and glycolytic reserve (*Figure 4D*; *Papandreou et al., 2006*). This short-term exposure to hypoxia at the time of activation also led to increased baseline oxygen consumption rates, increased spare respiratory capacity, and ATP production (*Figure 4E*). Non-glycolytic acidification, non-mitochondrial respiration, and proton leak were not impacted by low oxygen conditioning (*Figure 4—figure supplement 1I and J*). The change in oxygen consumption rates was accompanied by both an increased mitochondrial DNA copy number (*Figure 4—figure supplement 1K*) and increased Mitotracker (indicating increased mitochondrial mass) and increased TMRM staining (indicating a shift in mitochondrial membrane potential) (*Figure 4F*) when compared to cells continuously grown in 21% $O_2$, as measured at 7 d following initial activation.

To better understand the metabolic adaptation adaptations of CD8$^+$ T cells activated in 1% $O_2$ for a single day, we performed mass spectrometry analysis of several metabolites involved in T cell metabolism including glucose, amino acids, and acylcarnitines (*Figure 4—figure supplement 1L*). Analysis of metabolites feeding the TCA cycle showed that 1% 1 d-treated CD8$^+$ T cells accumulate acylcarnitines (namely, laurylcarninite, oleoylcarnitine, palmitoylcarninite, and mysteroylcarnitine) (*Figure 4G*). Acylcarnitines are intermediates in fatty acid metabolism that are involved in the transport of fatty acids into the mitochondria for energy production (*McCoin et al., 2015*). Their accumulation suggests that the increased oxygen consumption rates displayed by hypoxia-conditioned T cells may result from an increased contribution of these lipids to oxidative phosphorylation (*Figure 4E*).

It has been previously established that hypoxia rapidly modulates histone methylation and chromatin structure (*Batie et al., 2019*; *Chakraborty et al., 2019*). In line with this, our study reveals that even after 6 d of conditioning in ambient oxygen levels, exposure to 1% $O_2$ during T cell activation for a single day led to altered levels of specific post-translational histone modifications (*Figure 4H*).

The broad reduction in histone methylation could result from decreased methionine levels (*Figure 4I*). Methionine serves as a precursor for S-adenosylmethionine (SAM), the substrate for methyltransferase reactions: thus changes in methionine levels may impact epigenetic and epitranscriptomic regulation (*Yu et al., 2019*). While SAM levels were under the detection limit, analysis of metabolites involved in the methionine cycle showed that 5-methylthioadenosine (MTA) is reduced by short, low oxygen conditioning of T cells during activation (*Figure 4I*). MTA participates in the methionine salvage pathway, and a reduction in its levels could explain the drop in methionine levels

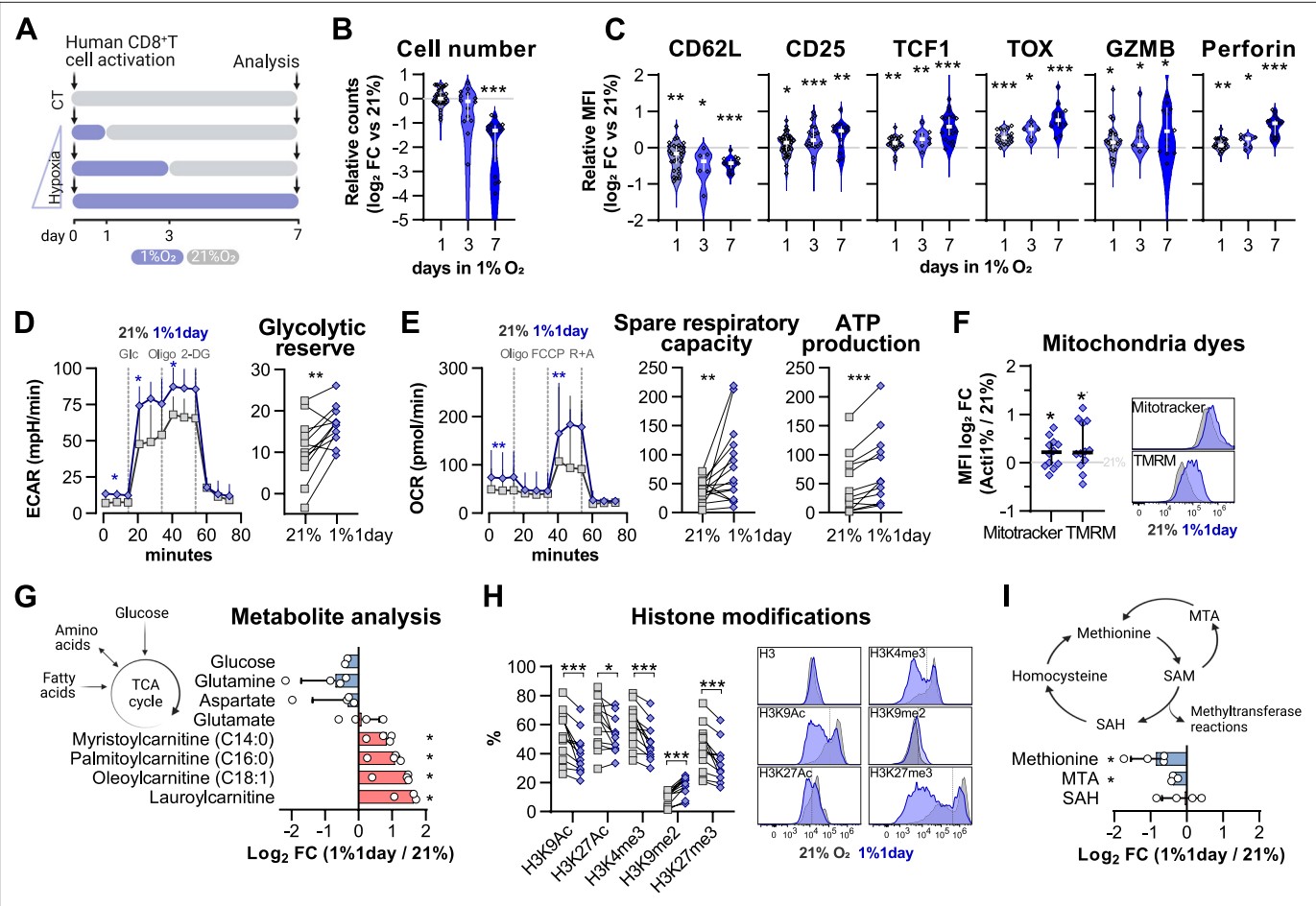

**Figure 4.** Impact of exposure to hypoxia in human CD8+ T cell differentiation and metabolism. (**A**) Human CD8+ T cells were activated in 1% O₂ for 1, 3, or 7 d. Cells continuously grown in 21% were used as control (CT), and all analyses were performed on day 7. (**B**) Number of CD8+ T cells after 7 d of increasing time of exposure to 1% O₂ as shown in (**A**). Cell counts assessed by flow cytometry using counting beads and presented as log₂ fold change (FC) relative to 21% cultures; n = 14–29. (**C**) Expression of differentiation markers shown as log₂ FC in median fluorescence intensity (MFI) relative to 21% CT (horizontal grey line) following conditioning to 1% O₂ as described in (**A**); n = 8–38. (**D, E**) Seahorse analysis of human CD8+ T cells exposed for 1 d to 1% O₂ (1% 1 d) as shown in (**A**). (**D**) ECAR trace and glycolytic reserve obtained with glucose stress test involving injection of glucose (Glc), Oligomycin (Oligo), and 2-DG; (**E**) OCR trace, spare respiratory capacity, and ATP production obtained with mitochondria stress test involving injection of oligo, FCCP, and rotenone + antimycin A (R+A); n = 18–24. (**F**) Signal of Mitotracker DeepRed and TMRM in 1% 1 d CD8+ T cells. Left: log₂ FC relative to 21% controls; right: representative histograms 21% control (grey) and 1% 1 d cells (blue); n = 12. (**G**) Mass spectrometry analysis metabolites feeding the TCA cycle including glucose, amino acids, and fatty acids. Shown as mean difference plots showing log₂ FC of metabolite levels in 1%1 d cells relative to 21% grown CD8+ T cells; n = 3–4. (**H**) Flow cytometry analysis of post-translational histone modifications in 1% 1 d and 21%T cells. Left: proportion of cells expression high levels of each histone modification. Right: representative histograms for 21% controls (dark grey) and 1% 1 d cells (blue); includes total histone 3 (H3) expression profile and dotted line defines the gate for each marker; n = 12. (**I**) Mass spectrometry analysis of metabolites involved in the methionine cycle including methionine, S-adenosylhomocysteine (SAH), and 5-methylthioadenosine (MTA). S-Adenosyl methionine (SAM) and homocysteine were under detection limit. Results are shown as median ± IQR (expect panel **G, H** and **J**) or as before-after plots. Each data point represents an independent donor. *p<0.05, **p<0.01, ***p<0.001; Wilcoxon matched-pairs signed-rank test (**B–E, H**), and one-sample *t* test (**F, G, I**).

The online version of this article includes the following source data and figure supplement(s) for figure 4:

Source data 1. Raw data and detailed analysis for *Figure 4*.

Figure supplement 1. Effect of low oxygen conditioning protocols in human CD8+ T cell differentiation and metabolism.

Figure supplement 1—source data 1. Raw data and detailed analysis for *Figure 4—figure supplement 1*.

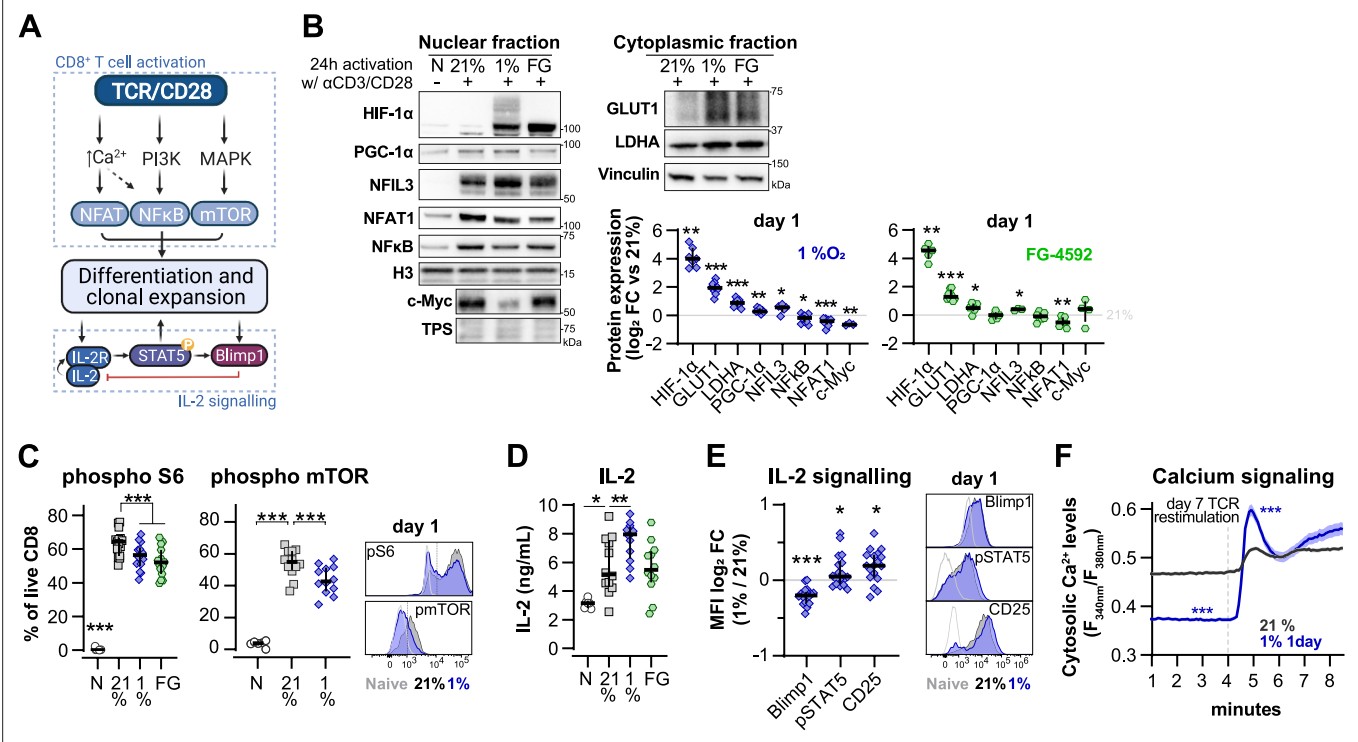

**Figure 5.** Effect of hypoxia and hypoxia-inducible factor (HIF) stabilisation in CD8+ T cell activation. (**A**) TCR signalling. (**B**) Protein analysis 24 hr after activation of human CD8+ T cells in 21% $O_2$, 1% $O_2$, or 21% $O_2$ + 50 µM FG-4592 (FG). Non-activated cells (N) were used as negative control. Histone 3, Lamin B, HDAC, or total protein stain (TPS) were used as loading control for nuclear fraction proteins, and vinculin was used as loading control for the cytoplasmic fraction. Left: representative immunoblots; right: log₂ fold change (FC) in protein expression of cells activated in 1% (top) or with FG (bottom) relative to 21% DMSO controls; n = 3–12. (**C**) Flow cytometry analysis of the proportion of cells positive for phospho-S6 Ribosomal Protein (Ser235/236, left) and phosphor mTOR (Ser2448, middle) for the same experimental conditions described in (**B**). Right: representative histograms for naive (light grey) and cells activated in 21% (dark grey) and 1% $O_2$ (blue); dotted line defines the gate for each marker; n = 6–15. (**D**) IL-2 ELISA performed on supernatant of cells activated for 1 d as described in (**B**); n = 6–12. (**E**) Flow cytometry analysis of Blimp1, phospho STAT5 (Y694, pSTAT5), and CD25 in cells activated in 1% $O_2$ for 24 hr. Left: median fluorescence intensity (MFI) log₂ FC relative to 21% $O_2$. Right: representative histograms for naive (light grey) and cells activated in 21% (dark grey) and 1% $O_2$ (blue). (**I**) OT-I T cells were activated for 24 hr in 1% $O_2$ with SIINFEKL and expanded for 6 d in 21% $O_2$ before single-cell cytosolic $Ca^{2+}$ was measured by microscopy using the radiometric dye Fura-2AM ($F_{340nm}/F_{380nm}$) in the presence of 1 mM $Ca^{2+}$ at day 7. Single-cell time-lapse imaging was performed at 5 s intervals with baseline fluorescence measured for 4 min, followed by another 6 min reading after addition of SIINFEKL. Number of cells analysed (21% control [grey], n = 249; 1% 1 d [blue], n = 128) are derived from multiple coverslips and two OT-I donor spleens; mean ± SEM. All results were pooled from at least two independent experiments and are shown as median ± IQR. Each data point represents an independent donor; *p<0.05, **p<0.01, ***p<0.001; one-sample *t* test (**B**), Šídák's multiple-comparisons test relative to 21% (**C**, **D**), Wilcoxon matched-pairs signed-rank test (**E**) and Mann–Whitney test (**F**).

The online version of this article includes the following source data and figure supplement(s) for figure 5:

**Source data 1.** Raw data and detailed analysis for *Figure 5*.

**Figure supplement 1.** Effect of hypoxia and hypoxia-inducible factor (HIF) stabilisation in CD8+ T cell activation.

**Figure supplement 1—source data 1.** Raw data and detailed analysis for *Figure 5—figure supplement 1*.

found in 1%1day CD8+ T cells (*Figure 4I*). These epigenetic alterations highlight the profound effects of oxygen tensions during T cell activation and may underlie the persistent effects of a single day of 1% $O_2$ on T cell differentiation and metabolism.

## Low oxygen tensions impact TCR signalling and enhance IL-2 production

To understand the impact of transient low oxygen tensions on T cell activation, we next characterised the expression of transcription factors and other proteins related to TCR and IL-2 signalling in CD8+ T cells at 24 hr following of activation in 1% $O_2$ (*Figure 5A*). We compared the alterations elicited by 1% $O_2$ to those seen during differentiation in ambient oxygen (21%) cultured T cells, and

to FG-4592-treated T cells (*Figure 5B* and *Figure 5—figure supplement 1A*). Nuclear translocation of NFAT1 was reduced by 1% $O_2$ and PHD inhibition, whereas the levels of nuclear factor kappa B (NFkB) were only reduced in the nuclear fraction of 1% $O_2$ activated T cells. Both NFAT1 and NFkB are crucial for T cell differentiation (*Serfling et al., 1995*). Levels of peroxisome proliferator-activated receptor gamma coactivator 1-alpha (PGC-1α, a master regulator of mitochondrial biogenesis; *Liang and Ward, 2006*) and c-Myc (a primary mediator of glycolysis in T cells; *Wang et al., 2011*) were respectively increased and decreased by T cell activation in 1% $O_2$ (*Figure 5B*).

The increase in HIF-1α levels following activation was augmented by exposure to 1% $O_2$ or PHD inhibition (*Figure 5B*). This correlated with increased levels of the known HIF targets glucose transporter 1 (GLUT1) and lactate dehydrogenase A (LDHA), as well as the Nuclear Factor Interleukin 3 Regulated (NFIL3) (*Figure 5B*), previously reported to be increased following ectopic expression of HIF-α or culture in 1% $O_2$ for 24 hr (*Ross et al., 2021*; *Veliça et al., 2021*). As NFIL3 has been linked to perforin production by NK cells, this transcription factor could underpin the increased T cell production of effector molecules following low oxygen conditioning or increased HIF signalling (*Rollings et al., 2018*). Moreover, 1% $O_2$ culture and PHD inhibition both decreased mTOR activation, as assessed by the reduction in phosphorylation of mTOR (Ser2448) and its target S6 Ribosomal Protein (Ser235/236) (*Figure 5C*). mTOR signalling is important for the growth and survival of activated T cells, and is also involved in the metabolic rewiring of T cells by driving the expression of c-Myc and HIF-1α (*Finlay et al., 2012*; *Wang et al., 2011*). Hypoxia and HIF have been previously shown to dampen mTOR signalling (*Wouters and Koritzinsky, 2008*), which could explain the above-mentioned reduced expression of c-Myc in T cells activated under low oxygen tensions.

T cell activation in low oxygen increased the production of IL-2 when compared to cells activated in ambient oxygen (*Figure 5D*). While FG-4592 treatment did not impact IL-2 production in T cells, we have previously found that *Il2* gene expression is increased by ectopic HIF-1α in mouse T cells (*Veliça et al., 2021*). In fact, we observed that the increase in IL-2 production following 24 hr of T cell activation in hypoxia was accompanied by an increase in STAT5 phosphorylation (Y694) and in the expression of IL-2 receptor subunit CD25 (*Figure 5E*). STAT5 is a positive regulator of IL-2 signalling previously shown to be induced by hypoxia in non-immune cells (*Joung et al., 2003*), whereas CD25 is one of the most consistently upregulated targets in T cells cultured under low oxygen tensions or following increased HIF signalling (*Figures 1A and 3B*). CD25 upregulation by hypoxia in T cells has also been shown by others (*Caldwell et al., 2001*; *Ross et al., 2021*) In contrast with pSTAT5 and CD25, expression of Blimp-1 was decreased in CD8$^+$ T cells following 24 hr of activation in 1% $O_2$ (*Figure 5E*). This could underpin the observed increase in IL-2 production and PGC-1α expression in T cells activated in low oxygen (*Figure 5B and D*) given that Blimp-1 is a transcriptional repressor of IL-2 signalling and PGC-1α (*Malek, 2008*; *Scharping et al., 2021*). Blimp-1 expression has been shown to be elicited both by T cell activation and HIF signalling (*Chiou et al., 2017*; *Martins and Calame, 2008*; *Veliça et al., 2021*). Accordingly, while we found that continuous T cell culture in low oxygen drives Blimp-1 expression (*Figure 5—figure supplement 1B*), the observed reduced expression following 24 hr of T cell activation in 1% $O_2$ might result from the concomitant decreased nuclear translocation of NFAT1 and NFkB.

Another aspect of T cell activation that has been linked with hypoxia is calcium signalling (*Neumann et al., 2005*). We performed single-cell fluorescent microscopy using the ratiometric calcium dye Fura-2AM, and OT-I cells reactivated with cognate antigen SIINFEKL (*Figure 5F*). Differentiation of OT-I cells in response to short 1% $O_2$ exposure during activation was similar to that of human CD8$^+$ T cells (*Figure 5—figure supplement 1C*). We found that conditioning to 1% $O_2$ exclusively during the first 24 hr of activation of OT-I cells resulted in lower baseline intracellular calcium levels on day 7 when compared to cells continuously cultured in ambient oxygen. However, restimulation with SIINFEKL at day 7 induced a much higher store-operated calcium entry (SOCE) in the cells conditioned to 1% $O_2$ (*Figure 5F* and *Figure 5—figure supplement 1D*). Activation of SOCE and the concomitant rise in cytosolic calcium is a key event in TCR signalling, leading to activation of calcineurin and translocation of nuclear factor of activated T-cells (NFAT) to the nucleus (*Serfling et al., 1995*); indeed, we do see an accompanying increase in NFAT1 nuclear translocation in hypoxia-conditioned group at day 7 (*Figure 5—figure supplement 1E*).

These findings suggest that low oxygen levels at the time of activation specifically shape T cell differentiation. Most, but not all, alterations elicited on T cell activation signalling by low oxygen

conditioning can be recapitulated by PHD inhibition, indicating that hypoxia drives HIF-dependent and HIF-independent reprogramming of T cells at the time of T cell activation.

## Short hypoxic conditioning improves T cell function against solid tumours

Lastly, we assessed the effect of the duration of low oxygen stimulation on CAR-T cell function. For this, we conditioned HER2- and CD19-CAR-T cells with low oxygen tensions for 1 d (*Figure 6A*) or 3 d (*Figure 6—figure supplement 1A*) during a 7 d culture protocol. Transduction efficacy of CAR vectors was not impacted by the culture strategies (*Figure 6—figure supplement 1B*). Cytotoxic CAR-T cell function was not improved by the more prolonged 3 d exposure to 5 or 1% $O_2$ (*Figure 6—figure supplement 1C–F*). However, a single day of 1% $O_2$ conditioning followed by 6 d of culture in 21% enhanced the killing capacity of both HER2- and CD19-CAR-T cells, with a significantly increased secretion of IFN-γ (*Figure 6B and C* and *Figure 6—figure supplement 1G and H*). Conditioning CAR-T cells for 1 d at 5% $O_2$ did not impact cytotoxic function (*Figure 6—figure supplement 1I and J*). As peripheral blood mononuclear cells (PBMCs) are used to generate CAR-T cells in the clinic, we confirmed that similar to HER2 CAR-T arising from isolated CD8$^+$ T cells, PBMC HER2 CAR-T cells conditioned for 1 d to 1% $O_2$ and expanded for 6 d in ambient oxygen also displayed enhanced killing capacity when compared to 21% controls (*Figure 6—figure supplement 1K*).

Adoptive transfer of HER2 CAR-T cells to NSG mice bearing HER2$^+$ SKOV3 tumours was used to functionally compare CAR-T cells exposed for 1 d to 1% $O_2$, to cells continuously cultured in 21% $O_2$ for 7 d prior to ACT (*Figure 6D*). We found that a short 1% $O_2$ conditioning ex vivo during activation improved CAR-T expansion in vivo, as assessed by analysis of the blood from recipient animals 7 d after ACT (*Figure 6E*). Additionally, 1% conditioned CAR-T cells were found to have a higher proportion of effector memory cells (CD45RO$^+$CCR7$^-$) in peripheral blood when compared to 21% oxygen cultured CAR-T cells (*Figure 6F*). A single day of 1% $O_2$ conditioning of HER2-CAR-T cells significantly enhanced their immunotherapeutic function against SKOV3 tumours, as shown by significantly decreased tumour growth rates, and by increased animal survival, relative to animals receiving 21% $O_2$-treated CAR-T cells (*Figure 6G and H*).

To better understand the enhanced antitumour function of T cells activated under low oxygen tensions, we conducted a tumour infiltration experiment with OT-I cells activated for 1 d at 1 or 21% $O_2$, and expanded for 6 d in ambient oxygen prior to injection in B16-OVA tumour-bearing animals (*Figure 6I*). We confirmed that similarly to human CAR-T cells, 1% 1 d OT-I cells show enhanced antitumour function in vitro when compared to 21% controls (*Figure 6—figure supplement 1L*). We additionally found that longer exposure to 1% $O_2$ could hinder antitumour T cell function (*Figure 6—figure supplement 1L*). The use of the CD45.1 and CD45.2 markers allowed us to follow transfused cells after adoptive transfer into experimental recipients and tumour sizes were similar between experimental groups (*Figure 6—figure supplement 1M and N*). Following ACT into tumour-bearing hosts, we found a striking increase of OT-I infiltration in the 1% 1 d group relative to the 21% controls, both per gram of tumour and per million CD45$^+$ cells (*Figure 6J* and *Figure 6—figure supplement 1O*).

Overall, our data shows that a short period of low oxygen conditioning during early T cell activation partially impairs T cell activation signalling, which results in a permanent shift in T cell differentiation, metabolism, epigenetic markers, and improved antitumour T cell function. While low oxygen conditioning in the last 3 d of T cell culture did not improve T cell function relative to 21% controls, continuous ex vivo T cell expansion in 1% $O_2$ impaired T cell function. These findings, which are summarised in *Figure 7*, indicate that oxygen tensions during activation can act to permanently shift T cell function and that the hypoxic response can be harnessed to improve T cell-based immunotherapeutic strategies.

## Discussion

Research investigating the role of hypoxia in CD8$^+$ T cell-mediated immunity has generally focused on the immunosuppressive consequences of oxygen depletion in the tumour microenvironment (*Chouaib et al., 2017*; *Jayaprakash et al., 2022*). While tumour hypoxia correlates with poor patient survival and reduced T cell infiltration (*Bertout et al., 2008*; *Hatfield et al., 2015*), data presented here show that the effects of low oxygen tensions on CD8$^+$ T cells are not solely inhibitory and can significantly

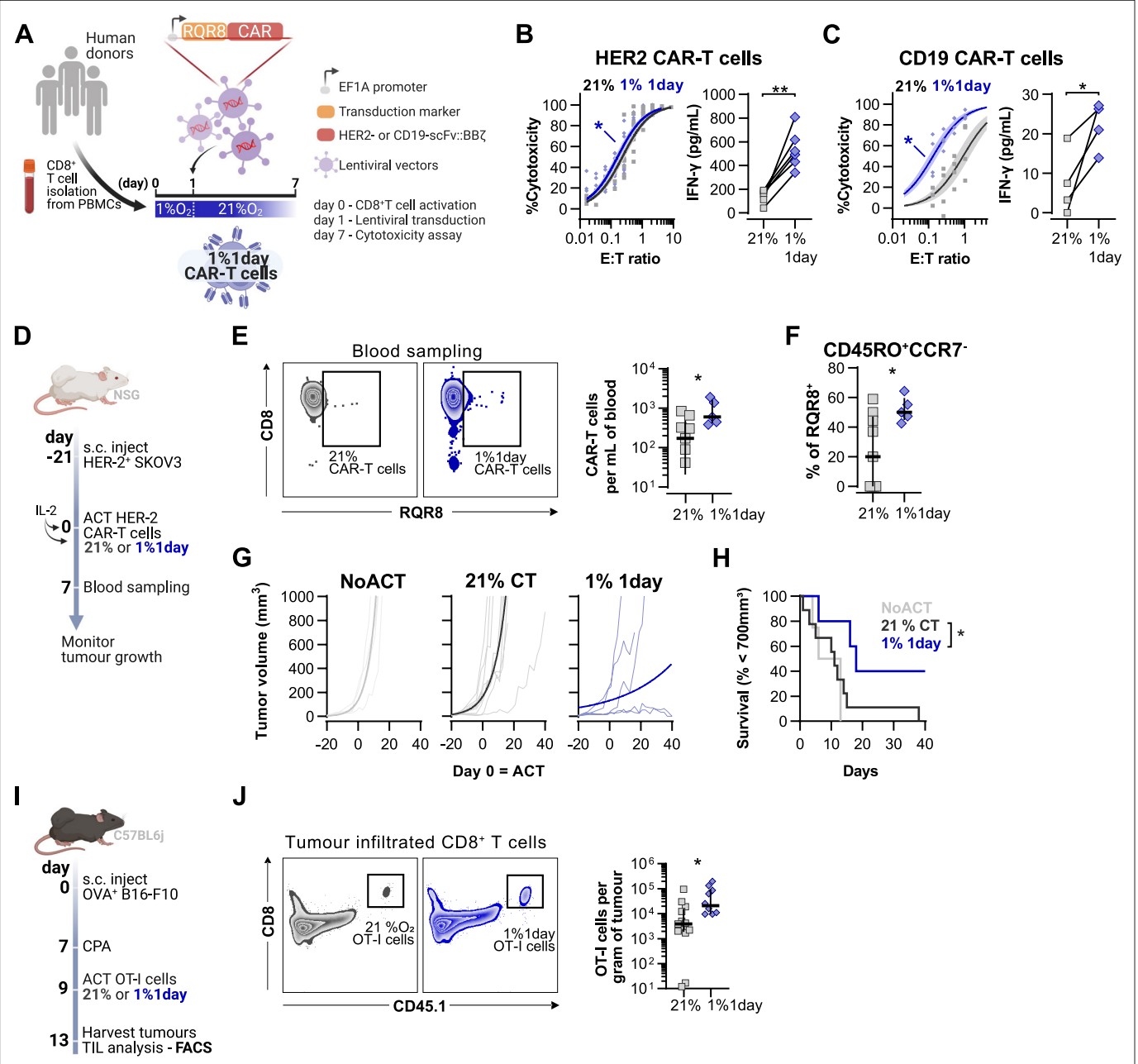

**Figure 6.** Effects of short hypoxic conditioning in T cell function against solid tumours. (**A**) Generation of CAR-T cells. Human CD8+ T cells were activated, transduced with HER2- or CD19-CAR vectors the day after and assayed on day 7. Cells were activated in 1% $O_2$ for 24 hr and swapped to 21% until day 7. CAR-T cells continuously maintained in 21% $O_2$ were used as controls and RQR8 was used as transduction marker. (**B**) In vitro cytotoxicity assay with HER2 CAR-T cells cultured according to (**A**). CAR-T cells were co-cultured with SKOV3 tumour cells at different effector:target (E:T) ratios and cytotoxicity was assessed after 14–18 hr of co-culture at 1% $O_2$. Left: dose–response curves (plotted with 95% confidence intervals represented in shaded areas) determined with non-linear regression ([agonist] vs. normalised response); right: $EC_{50}$ values obtained from non-linear regression; n = 6. (**C**) In vitro cytotoxicity assay with CD19 CAR-T cells cultured according to (**A**). CAR-T cells were co-cultured with RAJI tumour cells at a E:T ratio of 2:1 and cytotoxicity was assessed after 14–18 hr of co-culture at 5% $O_2$. Left: cytotoxicity relative to cells transduced with vector control containing RQR8 alone; right: IFN-γ ELISA using supernatants of the co-culture between CAR-T cells and RAJI; n = 4. (**D**) Model of CAR-T cell therapy. NOD. Cg-Prkdc^scidIl2rg^tm1Wjl/SzJ mice were inoculated with SKOV3 cells and injected with HER2 CAR-T cells generated according to (**A**). Peripheral blood was sampled and analysed by flow cytometry. Tumour growth was monitored every 2–3 d until day 40. (**E**) Blood analysis on day 14. Left: representative flow cytometry plots with events pre-gated on live, singlet, human CD45+. Right: frequency of adoptively transferred CAR-T cells per millilitre of peripheral blood; n = 5–7. (**F**) Percentage CD45RO+CCR7- cells among RQR8+ cells in peripheral blood on day 7 after ACT; n = 5–7. (**G**) Tumour growth curves. Animals not receiving T cells (NoACT) were used as negative controls. Thin lines: individual animals; thick lines: Malthusian growth curve fit; n = 5–12.

*Figure 6 continued on next page*

*Figure 6 continued*

(**H**) Survival curves for tumour growth shown in (**F**). Threshold for survival was set at 700 mm$^3$. (**I**) Tumour infiltration model. C57BL/6j mice were injected subcutaneously with OVA-expressing B16-F10 tumour cells and 7 d later were lymphodepleted with CPA. Mice bearing tumours for 9 d were then intraperitoneally injected with OT-I cells continuously expanded for 7 d in 21% or conditioned to 1% $O_2$ during the first 24 hr of TCR stimulation. Tumours were harvested on day 13 and processed to single-cell suspensions for flow cytometric analysis of tumour-infiltrated lymphocytes (TIL). Endogenous and adoptive populations of TILs were distinguished by the allelic variants of CD45. (**J**) Representative flow cytometry plots pre-gated on live, singlet, CD45+ events (left) and OT-I cells per gram of tumour (right); n = 10–12. *p<0.05, **p<0.01, ***p<0.001; Wilcoxon matched-pairs signed-rank test relative to control (**B**), Mann–Whitney test relative to control (**C, E, F**) and log-rank (Mantel–Cox) test relative to 21% (**H**).

The online version of this article includes the following source data and figure supplement(s) for figure 6:

**Source data 1.** Raw data and detailed analysis for *Figure 6*.

**Figure supplement 1.** Evaluation of different protocols of low oxygen conditioning in CAR-T cell function.

**Figure supplement 1—source data 1.** Raw data and detailed analysis for *Figure 6—figure supplement 1*.

augment aspects of adaptive immune responses. Our data indicate that transitory hypoxia during antigen recognition can differentially prime T cells and give rise to enhanced antitumour activity.

In T cells, cognate antigen stimulation of the T cell receptor (TCR) triggers an acute HIF-1α upregulation (even at the high oxygen tensions present in tissue culture), which drives a transcriptional program that supports both effector differentiation and aerobic glycolysis (*Finlay et al., 2012*; *Nakamura et al., 2005*; *Palazon et al., 2017*). In CD8$^+$ T cells, HIF-1β and HIF-1α (but not HIF-2 α) deletion resulted in incomplete effector differentiation and reduced expression of glycolytic genes (*Clever et al., 2016*; *Doedens et al., 2013*; *Finlay et al., 2012*; *Liikanen et al., 2021*; *Palazon et al., 2017*; *Veliça et al., 2021*). Deletion of various negative HIF regulators, the HIF inhibitor (FIH/HIF1AN) or the von Hippel–Lindau disease tumour suppressor (VHL) protein, or the prolyl hydroxylase domain-containing proteins (PHD), or overexpression of modified, negative regulator-insensitive HIF-α proteins, exacerbates the terminal effector program (*Clever et al., 2016*; *Doedens et al., 2013*; *Finlay et al., 2012*; *Liikanen et al., 2021*; *Palazon et al., 2017*; *Veliça et al., 2021*). In adoptive cell transfer experiments, there are clearly complexities found in the role of HIF in overall T cell function: HIF-1α deletion and overexpression of VHL- and FIH-insensitive HIF-α impaired CD8$^+$ T cell antitumour

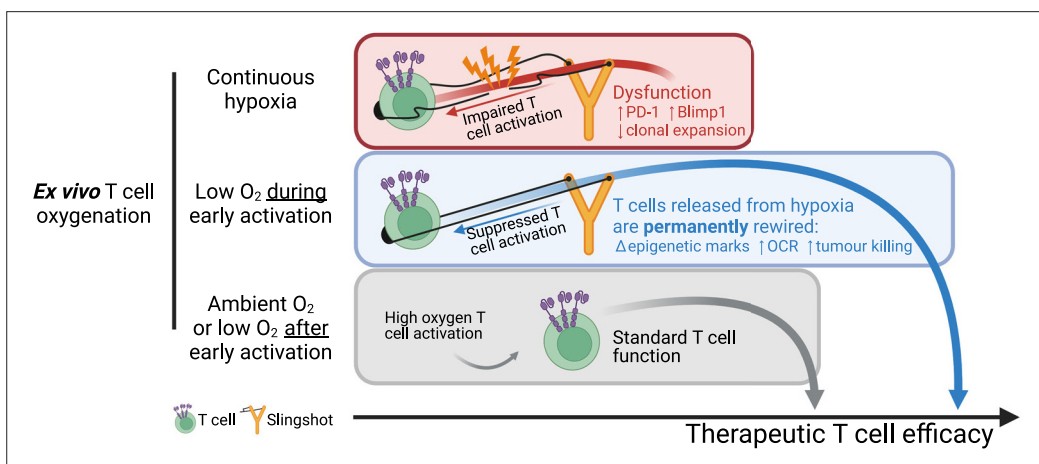

**Figure 7.** Summary of manuscript findings. Ex vivo culture of T cells in low oxygen continuously impairs T cell growth, increases expression of checkpoint receptor PD-1 and increased expression of IL-2 signalling repressor Blimp1. On the other hand, low $O_2$ conditioning exclusively during early activation (first 24 hr in 1% $O_2$) supresses T cell activation signalling but results in a permanent shift in histone modifications, increased OCR and ECAR, and increased antitumour function. These effects elicited by low oxygen condition include HIF-dependent and HIF-independent mechanisms such as altered NFAT and PGC-1α expression, respectively. T cells conditioned in low oxygen after early activation (day 3 onwards) are functionally similar to cells continuously grown in 21%. Altogether, these data indicate that while prolonged exposure to 1% $O_2$ is immunosuppressive, the temporary repression of T cell activation elicited by a short low oxygen conditioning functionally benefits T cells after they are released from the initial hypoxia-induced T cell repression.

function, while VHL deletion and overexpression of VHL-insensitive and FIH-sensitive HIF-2α improved CD8+ T cytotoxicity (*Liikanen et al., 2021*; *Palazon et al., 2017*; *Veliça et al., 2021*).

CD8+ T cell activation mediated an increase in HIF signalling that we found to be amplified by lower oxygen tensions. Ex vivo low oxygen conditioning of CD8+ T cells caused an increase in the production of effector molecules and cytotoxic function, but also severely impaired cell proliferation and increased cell death. This coupling of increased HIF-1α, increased cytotoxic function, and decreased cell growth is a finding which is consistent with previous reports (*Gropper et al., 2017*; *Ross et al., 2021*; *Xu et al., 2016*). Having previously shown that HIF-α deletion impairs CD8+ T cell function (*Palazon et al., 2017*), here we describe that HIF-1α accumulation, achieved through pharmacological PHD inhibition via FG-4592 or silencing of VHL, can boost antitumour CD8+ T cell function. Pharmacological inhibition of PHDs ex vivo has been shown to improve function in DMOG-treated CD4+ T cells (*Clever et al., 2016*). Here we show that similar to conditioning CD8+ T cells to low oxygen tensions before ACT, a transient boost in HIF signalling through ex vivo treatment with the PHD inhibitor FG-4592 improved the expansion and cytotoxicity of adoptively transferred tumour-specific mouse and human CD8+ T cells. This argues that oxygen tensions and/or HIF levels during T cell priming can have long-lasting functional effects on CD8+T cells.

Perhaps most surprisingly, we show that while long periods of low oxygen conditioning strongly reduced T cell expansion, the observed oxygen-dependent increases in effector function occurred regardless of whether the cells were cultured for short (1 d) or long periods of hypoxia (7 d) as long as the hypoxic exposure was coincident with activation. This is best illustrated by the fact that a single day of low oxygen conditioning during T cell activation, followed by expansion in higher oxygen tensions, was sufficient to alter T cell differentiation and metabolism, and to improve T cell infiltration of tumours and immunotherapeutic efficacy following adoptive T cell transfer. A recent study suggests standard cell culture conditions can lead to local hypoxia through a combination of cellular respiration and poor oxygen diffusion (*Tan et al., 2022*). Following 24 hr of 1% $O_2$ stimulation and 6 d of ambient oxygen T cell culture, we did not find considerable HIF-1α accumulation, and increasing media changes did not impact T cell differentiation. This suggests these cells under our conditions do not experience local hypoxia during ambient oxygen expansion, and that the only low oxygen experienced in our conditions occurs during T cell activation.

Although low oxygen conditioning, FG-4592 treatment, and VHL silencing enhanced T cell function and/or expansion in vivo, the distinct modes of inducing HIF signalling resulted in discrepancies in T cell differentiation. These can arise from additional prolyl hydroxylases apart from PHDs that can be inhibited by FG-4592 (*Maxwell and Eckardt, 2016*) and by the fact that VHL silencing is only induced by transduction of shVHL vectors 24 hr after T cell activation, thus missing the early T cell activation phase we feel are crucial for hypoxia-related improved T cell function. Additionally, shVHL vectors elicited a modest 25% reduction in VHL expression which, despite being sufficient to stabilise HIF-1α, could explain some of the discrepancies. Perhaps more importantly, hypoxic responses can be orchestrated by HIF, but also include HIF-independent inductions of other transcription factors such as PGC-1α (*Arany et al., 2008*) which have immunomodulatory roles. PGC-1α can be activated by the AMP-activated protein kinase (AMPK), which acts as a cellular energy sensor activated under conditions of metabolic stress, such as that caused by hypoxia, and can induce PGC-1α expression to promote mitochondrial biogenesis and oxidative metabolism (*Zong et al., 2002*). We have also found differences in survival of tumour-bearing animals receiving mouse and human T cells, and conditioned with low oxygen or FG treatment. These differences might be attributed to the duration of hypoxia and FG treatments, and to the different tumour models used. Despite these differences, we demonstrated that when compared to 21% $O_2$ cultured control cells, mouse and human CD8+ T cells cultured in 1% $O_2$ during T cell activation respectively display reduced B16-OVA tumour burdens, and extend survival of SKOV3 tumour-bearing animals.

24 hr of TCR stimulation and co-stimulation in 1% $O_2$ profoundly altered T cell activation. This short period of hypoxia at the time of activation reduced nuclear levels of NFAT1, NFkB, and c-Myc; decreased mTOR signalling; increased expression of HIF-1α, GLUT1, LDHA, NFIL3, and PGC-1α; and upregulated IL-2 signalling by increasing IL-2 production, CD25 expression, and STAT5 phosphorylation while reducing Blimp1 protein levels. After 6 d of ambient oxygen culture, T cells that had been conditioned to low oxygen for the first 24 hr of activation showed increased mitochondria biomass, enhanced basal and maximal glycolytic and respiratory rates, increased levels of lipid metabolism

intermediates and heightened SOCE following TCR restimulation. The underlying mechanisms for these changes may be attributed to the substantial alterations in the epigenetic histone signature elicited by brief hypoxic conditioning during T cell activation. While hypoxia-induced epigenetic modifications have been observed in other cell types (*Batie et al., 2019*; *Chakraborty et al., 2019*), their relevance to T cell functions remains incompletely understood. The present findings provide evidence for a previously unrecognised role for oxygen in the process of T cell activation and in the sustained modification of CD8[+] T cell differentiation. Further investigation is clearly required to elucidate the precise mechanisms by which hypoxia regulates T cell function.

The microenvironment, and the amount of available oxygen, at the time of antigen recognition thus has a highly persistent and determinative effect on T cell function, and this likely represents a novel and important aspect of T cell programming that requires further study. Many of these changes are HIF-driven, but some may well be related to other immunometabolic determinants present in the microenvironment of antigen recognition and activation. The potential for utilising these findings for short and easily achievable alterations in how therapeutic T cells are cultured presents an important new opportunity for improving efficacy in immunotherapy.

## Materials and methods

### Human material

Human blood samples from healthy male or female donors were obtained with written informed consent from the Cambridge Bioscience National Health Service Blood and Transplant at Addenbrooke's Hospital (Cambridge, UK) or from Karolinska University Hospital (Stockholm, Sweden); in both cases, informed consent and consent to publish forms part of the donation protocol. All blood was received as anonymised human donor material, with no identifiers attached to the donated material received. All relevant guidelines for Sweden and the UK for donated, anonymised human material were adhered to. For UK materials, ethical approval was obtained from the East of England-Cambridge Central Research Ethics Committee (06/Q0108/281); anonymised materials in Sweden did not require approval if used according to guidelines.

### Murine experiments

All animal experiments were approved by the regional animal ethics Committee of Northern Stockholm, Sweden (approval Dnr 19683-2021). C57BL/6J (CD45.2) animals were purchased from Janvier Labs, and NOD.Cg-*Prkdc*[scid]*Il2rg*[tm1Wjl]/SzJ were purchased from the Jackson Laboratory. Donor TCR-transgenic OT-I mice (JAX #003831, *Hogquist et al., 1994*) were crossed with mice bearing the CD45.1 congenic marker JAX #002014 (*Janowska-Wieczorek et al., 2001*).

### Cell lines

B16-F10 was originally purchased from ATCC (CRL-6475) and genetically modified to express ovalbumin, eGFP, and neomycin phosphotransferase (*Veliça et al., 2021*). The resulting ovalbumin-expressing B16F10 cells were cultured in DMEM high glucose with pyruvate (11995065 Thermo Fisher) containing 0.75 mg/mL G418 sulphate (10131027, Thermo Fisher). HEK293 was a gift from Prof. Dantuma (Karolinska Institute, Stockholm) and cultured in DMEM high glucose with pyruvate. SKOV3 was purchased from ATCC (HTB-77) and cultured in McCoy's 5A Medium (16600082, Thermo Fisher). Raji-GFP-Luc were purchased from Biocytogen (B-HCL-010) and cultured in complete RPMI (21875, Thermo Fisher). All media was supplemented with 1% penicillin streptomycin (15140122 Thermo Fisher) and 10% fetal bovine serum (FBS, A3160802 Thermo Fisher). Cell lines were frozen at low passage number (<5) in DMEM containing 10% DMSO and were typically passaged 3–4 times between thawing and experimental use. Cell line identities were confirmed by species-specific PCR analyses where relevant in non-human lines, as well as by functional properties. Otherwise cell lines were, as described above, obtained as authenticated lines from ATCC and were tested mycoplasma negative.

### T-cell isolation and activation and restimulation

Splenic murine CD8[+] T lymphocytes were purified with either positive and negative CD8 microbeads (130-117-044 and 130-104-075 Miltenyi, respectively) by magnetic-activated cell sorting (Miltenyi).

Activation was done in complete RPMI supplemented with 55 µmol/mL 2-ME (21985023, Gibco) and anti-mouse CD3/CD28 dynabeads (11453D, Thermo Fisher) at a 1:1 cell-to-bead ratio. Purified OT-I CD8+ T cells were activated with 0.1–1 µg/mL of the OVA-derived peptide SIINFEKL (ProImmune) or with anti-mouse CD3/CD28 dynabeads. CD8+ T cells were expanded in the presence of 10–50 U/mL recombinant human IL-2 (11147528001, Sigma). Human CD8+ T cells were purified from donor PBMCs by positive CD8 magnetic bead cell sorting (130-045-201, Miltenyi) and activated in complete RPMI supplemented with 30 U/mL IL-2 with anti-human CD3/CD28 dynabeads (11131D, Thermo Fisher) at a 1:1 cell-to-bead ratio. For experiments using PBMC CAR-T cells, 25 µL anti-human CD3/CD28 dynabeads were added to $2 \times 10^6$ total PBMCs in complete RPMI supplemented with 30 U/mL IL-2. After 3–4 d of the initial activation, dynabeads were removed and fresh media with IL-2 was added to T cell cultures. T cells were pre-incubated for at least 2 hr in the different oxygen tensions or with FG-4592 before activation and were cultured at a density of approximately $5–10 \times 10^5$ cells per mL and per $cm^2$. Low oxygen incubations were performed in a Ruskinn SCI-tive workstation.

## Flow cytometry

Single-cell suspensions were stained with Near-IR Dead Cell Stain Kit (Thermo Fisher) followed by surface and intracellular staining with fluorochrome-labelled antibodies (*Supplementary file 1*). Staining of cytoplasmic and nuclear antigens was performed using the Fixation/Permeabilization kit (BD Biosciences) and the Transcription Factor buffer set (BD Biosciences), respectively. To measure IFN-γ and TNF-α production, mouse OT-I T cells were incubated with RPMI supplemented with 1 µg/mL SIINFEKL with and treated with 5 µg/mL brefeldin 4 hr before intracellular staining and flow cytometry analysis. For proliferation assays, cells were loaded with CellTrace Violet (Thermo Fisher) according to the manufacturer's instructions. For mitochondria analysis, cells were loaded with 25 nmol/mL Mitotracker DeepRed (Invitrogen) or 30 nmol/mL TMRM (Invitrogen) for 20 min at 37°C. Samples were acquired in FACSCanto II (BD Biosciences) or in Aurora (Cytek Biosciences) flow cytometers and data analysed with FlowJo version 10. Transduced cells were sorted on an Aria III (BD Biosciences) following the surface antigen staining described above.

## Western blotting

Total cell pellets were lysed with urea-tris buffer (8 mol/mL urea, 50 mmol/mL Tris-HCl [pH = 7.5], 150 mmol/mL β-mercaptoethanol), sonicated twice for 45 s intercalated with 1 min incubation on ice and centrifuged at $14,000 \times g$, 4°C for 15 min. Histones were extracted with Histone Extraction Kit (ab113476, Abcam). Nuclear and cytoplasmic extracts from CD8+ T cells were obtained with the NE-PER Nuclear and Cytoplasmic Extraction Reagents (78833, Thermo Fisher). The Revert 700 Total Protein Stain (LI-COR) was used for analysis of total protein stain. Proteins (15–30 µg) were separated by SDS-PAGE and transferred to PVDF membranes before being probed with primary antibodies at a 1:1000 dilution (*Supplementary file 2*). Following the manufacturer's instructions, protein signal was detected using infrared-labelled secondary antibodies in an Odyssey imaging system (LI-COR) or using horseradish peroxidase-conjugated secondary antibody (R&D Systems) and ECL Prime (GE Healthcare) (imaged with an iBrightCL1000 (Thermo Fisher)).

## In vitro cytotoxicity assay

10,000 B16F10-OVA or SKOV3 cells were seeded per well in 96-well plates (flat bottom, Costar) and co-cultured for a minimum of 14 hr with varying ratios of mouse CD8+ OT-I or human CD8+RQR8+ HER2-CAR-T cells, respectively. Wells were washed twice with PBS to remove T cells, and the number of remaining target cells was determined by culturing with 10 µg/mL resazurin (Sigma) for 2 hr and measuring fluorescence signal in a plate reader. Cytotoxicity was calculated relative to wells with no T cells added (positive control) and to wells with no cancer cells added (negative control). 10,000 Raji cells were co-cultured with varying ratios of CD8+RQR8+ CD19-CAR-T cells for a minimum of 14 hr, and cytotoxicity was assessed by flow cytometry. The ratio of Raji cells to CountBright Absolute counting beads (Thermo Fisher) was used to calculate cytotoxicity. To determine specific cytotoxicity, data was normalised to the cytotoxicity of VC-transduced CD8+ T cells of the respective donor.

## Adoptive cell transfer experiments

8–15-week-old female C57BL/6j CD45.2$^+$ mice were inoculated subcutaneously with $5 \times 10^5$ B16-F10-OVA and conditioned 4 d later with peritoneal injection of 300 mg/kg cyclophosphamide (Sigma). On day 7, $0.5$–$1 \times 10^6$ CD45.1$^+$ OT-I CD8$^+$ T cells activated for 3 d with anti-CD3/CD28 mouse dynabeads in media with 10 U/mL IL-2 were peritoneally injected. 8–15-week-old female NOD.Cg-*Prkdc*$^{scid}$*Il2rg*$^{tm1Wjl}$/SzJ mice were inoculated subcutaneously with $1 \times 10^6$ SKOV3 and injected 21 to 7 d later with $5 \times 10^5$ RQR8$^+$ human CD8$^+$ CAR-T cells expanded for 7 d ex vivo. 100 U of IL-2 was peritoneally injected in NSG mice on the day of ACT and 3 d later. Animals were assigned randomly to each experimental group and tumour measurements were blinded. Tumour volume (a × b × b/2, where a is the length and b is the width) was measured every 2–3 d with electronic callipers until experimental end date as specified in figure legend.

## Vectors

DNA encoding a codon-optimised polycistronic peptide composed of RQR8 and anti-human HER2 (clone 4D5) or anti-human CD19 (clone FMC63) interspersed with picornavirus T2A and furin cleavage sequences was synthesised by GenScript. RQR8, used as the transduction marker, is a chimeric surface protein composed with domains from CD34 (for detection and purification with clone QBEND/10), CD8 (for anchoring at the cell surface), and CD20 (for depletion in vivo with anti-CD20 mAb rituximab) (*Philip et al., 2014*).

The generation of vectors coexpressing CAR molecules and an shRNA was possible due to the use of a microRNA scaffold. For expression of shRNAs, the commonly used RNA polymerase III promoters (e.g., U6), which can drive expression of small RNAs, often result in abundant production of small RNAs with heterogeneous 5′ ends; these can cause a deleterious increase of off-target effects. High levels of small RNA molecules can also become toxic to cells by saturating factors required for generating endogenous non-coding RNAs. To avoid these issues, shRNA-hairpins can be flanked with an optimised sequence of miR-30 (*Fellmann et al., 2013*) and can be expressed under the control of an RNA II polymerase promoter (e.g., EF1α). This promoter enables the coexpression of protein and shRNA, which are processed like endogenous miRNAs. MicroRNA-embedded shRNAs were generated as previously described (*Bofill-De Ros and Gu, 2016*). Briefly, 97-mer oligonucleotides (IDT Ultramers) coding for the respective shRNAs (*Moffat et al., 2006*) were PCR amplified using 10 µmol/mL of the primers miRE-XhoI-fw (5′-TGAACTCGAGAAGGTATATTG CTGTTGACAGTGAGCG-3′) and miRE-EcoRI-rev (5′-TCTCGAATTCTAGCCCCTTGAAGT CCGAGGCAGTAGGC-3′), 0.5 ng oligonucleotide template, and the Q5 High-Fidelity 2X Master Mix (NEB), and cloned HER2 CAR vectors containing the miRE scaffold sequence. All coding sequences were cloned into pCDCAR1 (Creative-Biolabs). Third-generation lentiviral transfer helper plasmids were obtained from Biocytogen. All sequences are available in *Supplementary file 3*.

## Seahorse analyses

Human CD8$^+$ T cells activated for 7 d were assayed in a Seahorse Extracellular Flux Analyzer XF96 (Agilent) as previously shown (*van der Windt et al., 2016*). $1.5$–$2 \times 10^5$ CD8$^+$ T cells were plated onto poly-D-lysine-coated wells in XF RPMI medium (Agilent) pH 7.4 supplemented with 2 mmol/mL glutamine. Cells used for mitochondria stress test were incubated in media supplemented with 10 mmol/mL glucose and sequentially injected with 1 µM oligomycin, 1.5 µM FCCP, and 100 nM rotenone +1 µM antimycin A. Cells used for glucose stress test were sequentially injected with 10 mmol/mL glucose, 1 µM oligomycin, and 50 mmol/mL 2-DG. All reagents were obtained from Sigma. A minimum of five technical replicates were used per biological replicate.

## Lentiviral transductions

For generation of lentiviral particles, $5 \times 10^6$ HEK293 cells were plated in 15 cm Petri dishes and transfected the day after with 50 µL FuGENE (E2311, Promega), 10 µg CAR-encoding vectors and 3.3 µg of each third-generation lentivirus helper vectors (CART-027CL, Creative-Biolabs). Supernatant media containing lentiviral particles was harvested 48 hr after transfection and used fresh or stored at –80°C. Lentiviral supernatants were spun onto non-treated 24-well plates, coated with 30 µg/mL Retronectin reused up to three times (T100B, Takara), at 2000 × *g* for 2 hr at 32°C and replaced with activated human CD8$^+$ T cells in fresh RPMI supplemented with 30 U/mL IL-2. Fresh media was added every 3 d.

## qPCR

Total RNA was extracted from isolated CD8$^+$ T cells (RNeasy kit, QIAGEN) and 300 ng of RNA were used for cDNA synthesis (First-Strand Synthesis kit, Invitrogen). All kits were used according to the manufacturer's instructions. Samples were run in technical duplicates. Quantitative real-time PCR (qPCR) was performed in a StepOnePlus system (Applied Biosystems) with 10 µL reactions composed of 4 µL cDNA (10× diluted), 1 µL primer solution (3 µmol/mL), and 5 µL FastStart Universal SYBR Green Master (4913914001, Roche). Hprt was used as housekeeping gene. Primers used were acquired from IdtDNA (Predesigned qPCR assays). Gene expression calculated with $2^{Ct(Hprt)-Ct(gene\ of\ interest)}$, where Ct values are the threshold cycles.

## Metabolite analysis

Cells were counted to determine viable cell numbers. 0.5–2 × 10$^6$ viable cells were harvested, washed with cold PBS, and metabolic activity quenched by freezing samples in dry ice and ethanol, and stored at −80°C. Before extraction, all samples were added with 10 µL of the internal standard solution, consisting of 200 µg/mL d4-succinic acid. Metabolites were subsequently extracted by addition of 1000 µL ice-cold LC-MS grade methanol to the cell pellets, followed by 30 min sonication. Sonication was carried out by adding ice to the ultrasound bath to maintain the temperature below 20°C. Following sonication, samples were centrifuged (12,000 × $g$, 20 min, 6°C) and 200 µL of the supernatant were transferred to LC-MS vials for the subsequent analysis. A targeted selected reaction monitoring (SRM) LC-MS/MS-based platform was used for metabolite relative quantification. Targeted analyses were performed on a Waters Acquity UPLC system coupled to a Xevo-TQ-S mass spectrometer (Waters, Milford, MA). An Acquity Premier BEH Amide Vanguard FIT column (2.1 mm × 100 mm, 1.7 µm), equipped with an integrated guard column, was used for the separation, with aqueous mobile phase consisting of 20 mmol/mL ammonium formate + 0.1% formic acid in double-deionised water and organic mobile phase consisting of 0.1% formic acid in acetonitrile. Column oven was set at 30°C, the injection volume was set to 2 µL, and the gradient was carried out at a flow rate of 0.4 mL/min. The following chromatographic gradient was used: 0 min, 95% organic; time range 0 → 1.5 min, 95% organic (isocratic step); time range 1.5 → 14.0 min, 95 → 55% organic (linear decrease); time range 14.0 → 14.2 min, 55 → 45% B (linear decrease); time range 14.2 → 16.5 min, 45% B (isocratic range); time range 16.5 → 17.0 min, 45 → 95% B (linear increase); time range 17.0 → 22.0 min, 95% B (isocratic column conditioning). MS analyses were performed using electrospray ionisation (ESI) operating in the positive and negative mode ionisation (independent chromatographic runs) using the following parameters: capillary potential (3.0 kV and –1.5 kV for positive and negative, respectively), source temperature 150°C, and desolvation temperature 550°C. Compound identification was based on retention time and, at least, one SRM matching with an in-house database adapted from *Medina et al., 2020*. The ion ratio between the transitions was used as an additional confirmation of the identification. For d4-succinate, the 121.0 > 58.1 transition was used.

## mtDNA analysis

Cells were harvested in 350 µL RLT lysis buffer and mixed in a volume of phenol/chloroform/isoamyl alcohol (25:4:1) (PCIAA). After mixing, the samples were centrifuged (16,000 × $g$ for 5 min), and 0.25–0.3 mL of the supernatant was mixed with 50 µg/µL glycoblue (AM9515, Thermo Fisher), 0.3 mol/mL sodium acetate, and 0.7× volume of isopropanol. After spinning samples at 16,000 × $g$ for 5 min at 4°C, DNA pellets were washed twice through addition of ice-cold 70% ethanol and centrifugation at 16,000 × $g$ for 10 min at 4°C. Air dried pellets (5–20 min) were then dissolved in 0.4 ml Tris–EDTA (TE) buffer. DNA was set to 2.5 ng/mL and analysed by qPCR using primers for nuclear 18S (FW 5'-TAGA GGGACAAGTGGCGTTC-3', RV 5'-CGCTGAGCCAGTCAGTGT-3') (*Thyagarajan et al., 2013*) and for mitochondrial DNA (FW 5'-GCCTTCCCCCGTAAATGATA-3', RV 5'TTATGCGATTACCGGGCTCT-3') (*Venegas and Halberg, 2012*).

## Single-cell calcium imaging

Frozen splenocytes from donor TCR-transgenic OT-I mice were acclimatised for 3 hr to either 21 or 1% O$_2$ before activation with 1 µg/mL SIINFEKL peptide for 48 hr. 24 hr post-activation, cells at hypoxia were moved to 21% and all cells were subsequently cultured at ambient oxygen until day 7. OT-I were kept in media supplemented with IL-2 (50 U/mL 0–48 hr, 20 U/mL 48 hr+). At day 7, OT-I T-cells

(~400,000 in 1 mL growth media) were loaded with 1 µmol/mL Fura-2 AM (F1221, Thermo Fisher) for 40 min room temperature (RT), in dark, on a horizontal rocker. Cells were washed and resuspended in Ringer's buffer (45 mmol/mL NaCl, 4 mmol/mL KCl, 10 mmol/mL glucose, 10 mmol/mL HEPES, 2 mmol/mL $MgCl_2$, 1 mmol/mL $CaCl_2$, pH 7.4) plus 20 U IL-2, before being left to de-esterify Fura-2 and adhere to poly-ornithine coated coverslips for at least 20 min. Coverslips were washed once more and mounted with 300 µL Ringer's solution + IL-2 on the microscopy set up (Zeiss Axiovert S100TV equipped with a pE-340fura (CoolLED) LED light source with LED 340 nm (excitation filter: 340/20) and 380 nm (excitation filter: 380/20) together with a T400 LP dichroic mirror and 515/80 emission filter, a sCMOS pco.edge camera and a Fluar ×20/0.75 objective). Ratiometric single-cell time-lapse imaging was performed at 5 s intervals with baseline fluorescence measured for 4 min, followed by another 6 min reading after addition of SIINFEKL (final concentration 4 µg/mL). VisiView 4.2.0.0 software (Visitron Systems) was used for data analysis, and calculated 340 nm/380 nm fluorescence ratios ($F_{340nm}/F_{380nm}$) were taken as directly proportional to cytosolic [$Ca^{2+}$].

## ELISA

To detect IFN-γ and IL-2 production, enzyme-linked immunosorbent assays (ELISA) were performed on T cell conditioned media. For this purpose, Human IFN gamma Uncoated ELISA kit (88-7316, Invitrogen) and IL-2 Human Uncoated ELISA Kit (88-7025, Invitrogen) were used according to the manufacturer's protocol and the signal was obtained in a microplate reader (Sunrise, Tecan Austria GmbH) at a wavelength of 450 nm.

## Analysis of tumour-infiltrated lymphocytes

8–15-week-old female C57BL/6j mice were inoculated subcutaneously with $1 \times 10^6$ B16-F10-OVA and conditioned 7 d later with a peritoneal injection of 6 mg CPA per animal (approximately 300 mg/kg). On day 10, mice were peritoneally injected with $1 \times 10^6$ OT-I $CD8^+$ T cells activated with anti-CD3/CD28 mouse dynabeads in media supplemented with 30 U/mL IL-2 and expanded ex vivo for 7 d in 21% $O_2$ or in 1% $O_2$ for 24 hr at the first 24 hr of activation. Animals were assigned randomly to each experimental group. On day 14, tumours were blindly processed in a GentleMACS dissociator (130-093-235, Miltenyi Biotec) using a Tumor Dissociation Kit (130-096-730, Miltenyi Biotec) and gentleMACS C tubes (130-093-237, Myltenyi Biotec). Infiltrated lymphocytes were analysed by flow cytometry with endogenous and adoptive populations being distinguished by the allelic variants of CD45.

## Statistics

Statistical analyses were performed with Prism 9 software (GraphPad). A p-value of $<0.05$ was considered significant, and the statistical tests and sample sizes (n) used are stated in figure legends. Normality tests and p-values are provided in source data files.

## Source data

Source data file for each figure and supplementary figure includes an Excel sheet for access to raw data, statistical analysis and cropped blots, as well as a folder for access to full unedited blots.

## Acknowledgements

The authors gratefully acknowledge the flow cytometry facility of the School of the Biological Sciences of the University of Cambridge for their support and assistance in this work. Bioenergetic experiments were performed at the Medical Research Council Toxicology Unit, University of Cambridge, Cambridge, UK. We acknowledge the Karolinska Institute Small Molecule Mass Spectrometry Core Facility (KI-SMMS), supported by KI/SLL, for support in sample analyses and scientific input. The authors thank Dr. Xiao-Ming Sun and Prof. Marion MacFarlane for providing access to the Seahorse XF96 Bioanalyser and helpful discussions. The authors would particularly like to acknowledge and thank Dr. Cristina M Branco, of Queen's University, Belfast, for helpful discussions, and for her role in the recruitment of, and facilitation of support for, Pedro P Cunha. The work was funded by the Knut and Alice Wallenberg Scholar Award, the Swedish Medical Research Council (Vetenskapsrådet 2019-01485), the Swedish Cancer Fund (Cancerfonden, CAN2018/808), the Swedish Children's Cancer Fund (Barncancerfonden PR2020-007), the Portuguese Foundation for Science and Technology scholarship

awarded to Pedro P Cunha (SFRH/BD/115612/2016), a Canadian Institutes of Health Research Fellowship to Brennan J Wadsworth and the Principal Research Fellowship (214283/Z/18/Z) to Randall S Johnson from the Wellcome Trust.

## Additional information

### Funding

| Funder | Grant reference number | Author |
|---|---|---|
| Wellcome Trust | 214283/Z/18/Z | Randall S Johnson |
| Knut och Alice Wallenbergs Stiftelse | Scholar | Randall S Johnson |
| Vetenskapsrådet | 2019-01485 | Randall S Johnson |
| Cancerfonden | CAN2018/808 | Randall S Johnson |
| Barncancerfonden | PR2020-007 | Randall S Johnson |
| Fundação para a Ciência e a Tecnologia | SFRH/BD/115612/2016 | Pedro P Cunha |
| Canadian Institutes of Health Research | Postdoctoral Fellowship | Brennan J Wadsworth |

The funders had no role in study design, data collection and interpretation, or the decision to submit the work for publication. For the purpose of Open Access, the authors have applied a CC BY public copyright license to any Author Accepted Manuscript version arising from this submission.

### Author contributions

Pedro P Cunha, Eleanor Minogue, Conceptualization, Formal analysis, Investigation, Methodology, Writing - original draft, Writing - review and editing; Lena CM Krause, Rita M Hess, David Bargiela, Brennan J Wadsworth, Investigation, Methodology; Laura Barbieri, Investigation, Methodology, Writing - review and editing; Carolin Brombach, Investigation; Iosifina P Foskolou, Formal analysis, Supervision, Investigation, Project administration, Writing - review and editing; Ivan Bogeski, Investigation, Methodology, Project administration; Pedro Velica, Conceptualization, Supervision, Investigation, Methodology, Project administration, Writing - review and editing; Randall S Johnson, Conceptualization, Formal analysis, Supervision, Funding acquisition, Writing - original draft, Project administration, Writing - review and editing

### Author ORCIDs

Pedro P Cunha ![ORCID] http://orcid.org/0000-0002-3814-6289
Lena CM Krause ![ORCID] http://orcid.org/0000-0003-3375-2850
Rita M Hess ![ORCID] http://orcid.org/0000-0001-5547-4172
Brennan J Wadsworth ![ORCID] http://orcid.org/0000-0002-9183-227X
Ivan Bogeski ![ORCID] http://orcid.org/0000-0001-9879-7174
Randall S Johnson ![ORCID] http://orcid.org/0000-0002-4084-6639

### Ethics

This study was performed in strict accordance with the recommendations for ethical experimentation in conformance with Swedish and EU laws and regulations. All of the animals were handled according to approved institutional animal care and use committee protocols. The protocol was approved by the Committee on the Ethics of Animal Experiments of the Ethical Review Board of Northern Stockholm and the Swedish Ministry of Agriculture. All surgery was performed under anaesthesia, and every effort was made to minimise suffering.

This study was performed in strict accordance with the recommendations for ethical experimentation in conformance with Swedish and EU laws and regulations. All of the animals were handled according to approved institutional animal care and use committee protocols. The protocol was approved by the Committee on the Ethics of Animal Experiments of the Ethical Review Board of Northern Stockholm

(approval Dnr 19683-2021) and the Swedish Ministry of Agriculture. All surgery was performed under anaesthesia, and every effort was made to minimise suffering.

### Decision letter and Author response

Decision letter https://doi.org/10.7554/eLife.84280.sa1
Author response https://doi.org/10.7554/eLife.84280.sa2

## Additional files

### Supplementary files

• Supplementary file 1. List of antibodies used for flow cytometry. Species reactivity: mouse (M), human (H), or both (M/H).

• Supplementary file 2. List of antibodies used for western blot analysis.

• Supplementary file 3. List of protein sequences and DNA sequences corresponding to miRNA-embedded shRNAs used in viral vectors.

• MDAR checklist

### Data availability

All data generated or analysed during this study are included in the manuscript and supporting file; Source Data files have been provided.

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
