## [Editor Report]

Your work employing solid preclinical models and in vitro experiments, and addressing how oxygen tension affects T-cell function, can help to improve cellular therapies for future patients.

---

## [Decision Letter]

**Decision letter after peer review:**

Thank you for submitting your article "Oxygen levels at the time of activation determine T cell persistence and immunotherapeutic efficacy" for consideration by *eLife*. Your article has been reviewed by 3 peer reviewers, and the evaluation has been overseen by me as the Reviewing Editor and W Kimryn Rathmell as the Senior Editor. The following individuals involved in the review of your submission have agreed to reveal their identity: Martin Böttcher (Reviewer #1); Helga Simon-Molas (Reviewer #2).

The reviewers have extensively studied your manuscript and all agree, that the topic you address is important to the field, that the findings might have broad implications, and the methods and data analysis are solid. You can find the detailed comments below, but all three reviewers independently addressed the discrepancies in the effects of genetic vs pharmacological modulation of HIF signalling (also in the murine vs human system) on T cell cytotoxicity and (anti-tumor) function. The current theoretical explanations of these discrepancies seem to be insufficient and will require additional experimental revisions.

*Reviewer #1 (Recommendations for the authors):*

Congratulations on this nice manuscript with important data to improve current therapeutic possibilities.

Although the data presented is in its entirety convincing there are some minor points I would like to be addressed:

1) Figure 4: It is important to clearly highlight/point out the time of analysis of the different shown panels. In its current form, it remains somewhat confusing and might lead to false conclusions by the reader.

2) Figure 4D/E: It seems that there occurred a problem with the Oligomycin used. While ECAR should increase (or at least remain stable) after Oligomycin treatment as compensation for lack of ATP by OxPhos, OCR after Oligomycin treatment should be reduced since the ATPase is inhibited and oxygen is not consumed anymore as the mitochondrial membrane potential/proton gradient is not released. Please carefully review the raw data and the used protocol for the Seahorse analysis and explain this logical error. Also, please provide a section of this analysis in Materials and methods.

3) Figure 4J: I would recommend showing also phospho-mTOR itself and not only a downstream protein.

4) Figure 5 B/C: Why were there different ways of depiction chosen? I would harmonize the data for both types of CART-cells, i.e. EC50 and IFN-γ, and also the depiction.

5) Figure 5E: The frequency of CART-cells seems rather low. Is there a logical explanation for this?

6) Figure 5D-E: It is very unfortunate that only the HER2-CART-cells were chosen in an in vivo model although also the CD19-CART-cells show better cytotoxicity/IFN-γ secretion after the pre-conditioning. If possible, a repetition using e.g. a lymphoma model would be favorable. However, I would also appreciate a discussion in case these additional experiments are not possible in the time frame of the revision.

*Reviewer #2 (Recommendations for the authors):*

While the data provided is consistent and analyzed in a high level of detail, some questions emerge from the data presented that could be further discussed. This is mostly related to the differences between the effects of low oxygen conditions and FG-4592 in vitro for T cell expansion (Figure 1) and for tumour volume and animal survival (Figure 2). Besides, CARs treated with FG or shVHL also have different outcomes in terms of cytotoxicity (Figure 3). Could the authors discuss this better?

Other important aspects to address:

– Figure 3D, E: the effects of FG and shVHL in cytotoxicity assays were performed at 1% oxygen. Is this because at 21% O2 there´s no HIF1a present and therefore the treatments have no effect? Did the authors perform the same experiments at 21% O2 pressure? In that case, it would be informative to include the data in the supplementary.

– Figure 4 D and E: D shows ECAR data and shows OCR data for a different Seahorse test. As it´s expected to be available from the assays, the corresponding data for OCR (D) and ECAR (E) should be shown in Supplementary. This would help understand better the effects of Oligomycin in these cells, which seem to not increase ECAR as would be expected, and also the effects of 2-DG on OCR, to see if the main source of TCA fuelling in these cells is glucose. Can the authors include these data?

– Similar to the assessment of memory effector cells in Figure 5F, did the authors explore that in ex vivo experiments? What´s the phenotype of T cells treated under low oxygen conditions during T cell stimulation, after the activation peak? Are there also more memory effector cells?

– The authors monitor the % of CAR-T cells in blood. Did they measure the % of TILs inside the subcutaneous tumor mass?

*Reviewer #3 (Recommendations for the authors):*

I see the importance and novelty of this work but due to the listed weaknesses in the public review, I would recommend the authors assess the O2 concentration experienced by T cells during their culturing protocols. Alternatively (or in addition), using the strategy of the above-cited study, lower cell medium volumes to improve O2 diffusion could be evaluated. Despite the fact that T cells are suspension cells they will descend to the bottom of the well and so could experience local hypoxia. More comprehensive metabolic characterisation comparing CAR-T cells following short hypoxic conditioning compared to no conditioning would add value. Discussion of potential HIF-independent mechanisms should also be discussed but important to assess them in follow-up work.

---

## [Author Response]

Reviewer #1 (Recommendations for the authors):Congratulations on this nice manuscript with important data to improve current therapeutic possibilities.Although the data presented is in its entirety convincing there are some minor points I would like to be addressed:1) Figure 4: It is important to clearly highlight/point out the time of analysis of the different shown panels. In its current form, it remains somewhat confusing and might lead to false conclusions by the reader.

To address this, we have now split this figure into two figures. Figure 4 now shows the different oxygen condition strategies and highlights the day 7 effects that result from 1% O_2_ conditioning during the first 24h of T cell activation. Figure 5 focuses on the relevance of oxygen tensions and HIF levels for T cell activation at 24 hours (Figure 5 B-E) and for T cell reactivation at day 7 (Figure 5F).

2) Figure 4D/E: It seems that there occurred a problem with the Oligomycin used. While ECAR should increase (or at least remain stable) after Oligomycin treatment as compensation for lack of ATP by OxPhos, OCR after Oligomycin treatment should be reduced since the ATPase is inhibited and oxygen is not consumed anymore as the mitochondrial membrane potential/proton gradient is not released. Please carefully review the raw data and the used protocol for the Seahorse analysis and explain this logical error. Also, please provide a section of this analysis in Materials and methods.

We thank the reviewer for calling our attention to this. Upon repeating the assay with 12 human donors and a new stock of oligomycin, we could confirm inhibition of ATP synthase does increases ECAR and reduce OCR in human CD8^+^T cells expanded for 7 days (Figure 4D-E and Figure 4 —figure supplement 1I-J). Seahorse analysis protocols are now part of the Materials and methods section.

3) Figure 4J: I would recommend showing also phospho-mTOR itself and not only a downstream protein.

We thank the reviewer and have now carried out this analysis. As expected by the decreased phosphorylation of S6, we were able to confirm that activation 1% O2 also decreases mTOR phosphorylation (Figure 5C).

4) Figure 5 B/C: Why were there different ways of depiction chosen? I would harmonize the data for both types of CART-cells, i.e. EC50 and IFN-γ, and also the depiction.

We have now harmonized the data for both types of CAR-T cells, and this can now be seen in Figures 6B and C. As seen in CD19-CAR cells, we have confirmed that HER-2 CAR-T cells conditioned to 1%O_2_ for 24h also secrete more IFN-γ following co-culture with target HER-2 cells.

5) Figure 5E: The frequency of CART-cells seems rather low. Is there a logical explanation for this?

Transduction efficiency for CAR experiments was 10% on average (Figure 6 —figure supplement 1B) which explains that the majority of the CD8^+^T cells in circulation are RQR8-. However, the transduction efficiency was similar between 21% controls and 1% 1day cells; following ACT and IL-2 administration in vivo, it is still clear that 1% 1day treated HER-2 CAR-T cells expanded to a greater extent in the blood (Figure 6E); and displayed an increase in effector memory cells (Figure 6F) when compared to 21% control treated HER-2 CAR-T cells.

6) Figure 5D-E: It is very unfortunate that only the HER2-CART-cells were chosen in an in vivo model although also the CD19-CART-cells show better cytotoxicity/IFN-γ secretion after the pre-conditioning. If possible, a repetition using e.g. a lymphoma model would be favorable. However, I would also appreciate a discussion in case these additional experiments are not possible in the time frame of the revision.

Whilst we show that the low oxygen conditioning protocol improves the function of different CAR-T cells (CD19- and HER-2 CAR-T cells) in vitro, we selected the latter for adoptive cell transfer experiments, as the target cells (SKOV3) for that CAR-T allow for the use of a preclinical solid tumour model; we felt such a model might be more relevant, given the greater difficulties inherent to targeting solid tumours with immunotherapy. Additionally, CD19 CAR-T cell therapy is currently established in the clinic against B cell malignancies, whereas we feel that improving the success of CAR-T cells targeted to solid tumours more strongly warrants innovative approaches and analysis. However, we have no reason to feel that the improvements shown here would not also be seen in lymphoma models.

Reviewer #2 (Recommendations for the authors):While the data provided is consistent and analyzed in a high level of detail, some questions emerge from the data presented that could be further discussed. This is mostly related to the differences between the effects of low oxygen conditions and FG-4592 in vitro for T cell expansion (Figure 1) and for tumour volume and animal survival (Figure 2). Besides, CARs treated with FG or shVHL also have different outcomes in terms of cytotoxicity (Figure 3). Could the authors discuss this better?

Whilst VHL silencing alterations to T cell function closely resemble those obtained with low oxygen conditioning (i.e., increased expression of effector molecules, and enhanced antitumour function in vitro and in vivo (Figure 3)), the same cannot be said for FG-4592 treatment. The in vitro effects of FG-4592 suggest this inhibitor might be immunosuppressive (since it reduces cell expansion (Figure 1F)) and may hinder CAR-T cell cytotoxicity (Figure 3D). However, FG-treated cells revealed enhanced adoptive cell transfer functions in vivo, indicating that the net long term effects of FG-4592 might potentiate T cell function. Relative to control T cells, FG-treated OT-I cells reduced tumour burdens and extended animal survival, and FG-treated CAR-T cells showed enhanced expansion in the blood following ACT and delayed tumour growth in 50% of the animals compared to 9% of the animals in the control T cell group. Nevertheless, finding different effects from VHL silencing vs FG-4592 reflects the complexity of the HIF pathway and its regulation. These points are now added to the 5^th^ paragraph of the Discussion section.

Other important aspects to address:– Figure 3D, E: the effects of FG and shVHL in cytotoxicity assays were performed at 1% oxygen. Is this because at 21% O2 there´s no HIF1a present and therefore the treatments have no effect? Did the authors perform the same experiments at 21% O2 pressure? In that case, it would be informative to include the data in the supplementary.

We have performed the cytotoxicity assays under different oxygen tensions. As shown in Figure 1 —figure supplement 1E, the cytotoxicity of low oxygen conditioned and FG-treated T cells was similar, regardless of the oxygen tension at which the co-culture cytotoxicity assay was performed. We now show cytotoxicity assays for FG-treated and VHL silenced CAR-T cells in 21%O_2_ (Figure 3 —figure supplement 1I).

– Figure 4 D and E: D shows ECAR data and shows OCR data for a different Seahorse test. As it´s expected to be available from the assays, the corresponding data for OCR (D) and ECAR (E) should be shown in Supplementary. This would help understand better the effects of Oligomycin in these cells, which seem to not increase ECAR as would be expected, and also the effects of 2-DG on OCR, to see if the main source of TCA fuelling in these cells is glucose. Can the authors include these data?

We have now provided new data generated with additional human donor T cells and obtained an overall canonical cellular response to ATP synthase inhibition with oligomycin (increased ECAR and decreased OCR). We also now provide more detailed analyses of these live metabolic assays (Figure 4 —figure supplement 1I-J) and show that 2-DG does not significantly impact OCR. This could indicate that additional carbon sources such glutamine or lipids might be fuelling the TCA cycle when glucose levels drop. Mass spectrometry analyses performed for this revision showed that acylcarnitines accumulate in 1% 1day cells (Figure 5G), which could indicate fatty acid oxidation might preferentially fuel the OCR. In future work we will perform Flex test assays to better understand the substrate dependency of low oxygen conditioned human T cells.

– Similar to the assessment of memory effector cells in Figure 5F, did the authors explore that in ex vivo experiments? What´s the phenotype of T cells treated under low oxygen conditions during T cell stimulation, after the activation peak? Are there also more memory effector cells?

We have confirmed low oxygen conditioning drives effector memory differentiation in a 7-day expansion protocol (Figure 4 —figure supplement 1H).

– The authors monitor the % of CAR-T cells in blood. Did they measure the % of TILs inside the subcutaneous tumor mass?

Given that the HER-2 CAR-T Act tumour model takes approximately 3 months to set up, and thus did not fit in the time frame of these revisions, we carried out an alternative analysis of the percentage of TILs found inside subcutaneous tumour masses, using the OT-I and B16-OVA model system. For this, we first confirmed that OT-I cells expanded for 7 days ex vivo phenotypically and functionally resemble human CAR-T cells when conditioned to low oxygen tensions (Figure 6 —figure supplement 1L). Longer exposure to 1%O_2_ drives stronger effector differentiation but also hinders antitumor T cell function. Following transfer of ACTs into tumour bearing hosts, we found a striking increase of OT-I infiltration in the 1% 1day group relative to the 21% controls: this indicates that low oxygen conditioning ex vivo can improve antitumour function in vivo, and correlates with an increased level of T cell infiltration and increased T cell expansion (Figure 6 I-J).

Reviewer #3 (Recommendations for the authors):I see the importance and novelty of this work but due to the listed weaknesses in the public review, I would recommend the authors assess the O2 concentration experienced by T cells during their culturing protocols. Alternatively (or in addition), using the strategy of the above-cited study, lower cell medium volumes to improve O2 diffusion could be evaluated. Despite the fact that T cells are suspension cells they will descend to the bottom of the well and so could experience local hypoxia. More comprehensive metabolic characterisation comparing CAR-T cells following short hypoxic conditioning compared to no conditioning would add value. Discussion of potential HIF-independent mechanisms should also be discussed but important to assess them in follow-up work.

We have now added discussion of this issue to the 4^th^ paragraph of the Discussion section. We also provide a more detailed metabolic analysis of low oxygen stimulated T cells with a mass spectrometry analysis of carbon sources utilized by T cells (Figure 4G and Figure 4 —figure supplement 1L). We found that 1% 1day T cells accumulate acylcarnitines, which could explain the observed increase in basal and maximum oxygen consumption rates resulting from short hypoxia stimulation during T cell activation (Figure 4E). Additionally, we now discuss additional HIF-independent mechanisms driven by hypoxic responses in the 5^th^ paragraph of the Discussion section.